nature
biomedical engineering

# CAR T cells expressing a bacterial virulence factor trigger potent bystander antitumour responses in solid cancers

Chuan Jin[1,2], Jing Ma[1,2], Mohanraj Ramachandran [1], Di Yu [1✉] and Magnus Essand [1✉]

Chimeric antigen receptor T cells (CAR T cells) are effective against haematologic malignancies. However, in solid tumours, their potency is hampered by local immunosuppression and by the heterogeneous expression of the antigen that the CAR targets. Here we show that CAR T cells expressing a pluripotent pro-inflammatory neutrophil-activating protein (NAP) from *Helicobacter pylori* trigger endogenous bystander T-cell responses against solid cancers. In mice with subcutaneous murine pancreatic ductal adenocarcinomas, neuroblastomas or colon carcinomas, CAR(NAP) T cells led to slower tumour growth and higher survival rates than conventional mouse CAR T cells, regardless of target antigen, tumour type and host haplotype. In tumours with heterogeneous antigen expression, NAP secretion induced the formation of an immunologically 'hot' microenvironment that supported dendritic cell maturation and bystander responses, as indicated by epitope spreading and infiltration of cytotoxic CD8+ T cells targeting tumour-associated antigens other than the CAR-targeted antigen. CAR T cells armed with NAP neither increased off-tumour toxicity nor hampered the efficacy of CAR T cells, and hence may have advantageous translational potential.

Therapies with adoptive transfer of autologous T cells engineered ex vivo with a chimeric antigen receptor (CAR) directed against CD19 are efficacious and approved for refractory and relapse B-cell leukaemia and lymphoma, while B-cell maturation antigen-directed CAR T-cell treatment is approved for multiple myeloma[1–4]. In solid tumours, a particular tumour-associated antigen that CAR T cells could target is often heterogeneously expressed, which impairs the efficiency of CAR T-cell therapy[5]. Furthermore, the immunosuppressive microenvironment in solid tumours obstructs CAR T-cell efficacy[6,7]. Therefore, combating CAR target antigen heterogeneity and reducing immunosuppression is of utmost importance in improving CAR T-cell therapy of solid tumours.

Various approaches to reinforce the efficacy of CAR T cells against solid tumours have been evaluated, including engineering CAR T cells to produce endogenous enzymes, cytokines, chemokines that potentiate either CAR T cells or other host immune cells, or both[8–15]. Many exogenous molecules of bacterial origin have strong immunomodulatory properties. One such protein is the neutrophil-activating protein (NAP) from *Helicobacter pylori*, which attracts innate immune cells, induces maturation and T helper type-1 (Th1) polarization of dendritic cells (DCs), and creates a local pro-inflammatory milieu with enhanced interleukin (IL)-12 production[16–18]. We investigate and report that arming of CAR T cells with NAP can improve their activity in various solid tumour models.

## Results

We use retroviral vectors (RV) (Fig. 1a) to engineer murine T cells. After validating inducible NAP expression (Supplementary Fig. 1), we evaluated the efficacy of CAR(NAP) T cells to target endogenously expressed murine CD19 and disialoganglioside (GD2) in two immunocompetent, syngeneic mouse models of cancer, A20

(lymphoma) and NXS2 (neuroblastoma) (Fig. 1b,c). Conventional CAR T cells and CAR(NAP) T cells exhibited similar cytotoxicity against both cell lines in vitro (Extended Data Figs. 1a,b and 2a–c), confirming that insertion of the NAP transgene did not compromise the cytotoxic capacity of CAR T cells. When evaluated in vivo, CAR(NAP) T cells controlled tumour growth and significantly prolonged the survival of tumour-bearing mice in both models (Fig. 1d–g, and Extended Data Figs. 1c–e and 2d–f). Approximately 75% of the mice became tumour-free when treated with CAR(NAP) T cells, while conventional CAR T-cell treatment only cured 30% of the mice with A20 lymphoma (Fig. 1d,e). Mice cured from A20 tumours were protected against rechallenge with the same tumour cell line (Fig. 1e), indicating the establishment of immunological memory. Decreased B-cell counts were observed in the blood of mice after treatment with CD19 targeting CAR(NAP) T cells but not in mice treated with conventional CAR T cells (Extended Data Fig. 1f), which is also a sign of potency of CAR(NAP) T cells. Of note, this enhanced efficacy of CAR(NAP) T cells was neither dependent on the nature of the target antigen, nor the tumour types or host haplotypes, as superior therapeutic efficacy was also observed for CAR(NAP) T cells in various tumour models, including NXS2-mCD19 neuroblastoma (Extended Data Fig. 3), Panc02-mCD19 pancreatic adenocarcinoma (Extended Data Fig. 4) and CT26-hPSCA colon carcinoma (Extended Data Fig. 5). These data confirm that CAR(NAP) T cells perform better than CAR T cells in vivo despite exhibiting similar potency in vitro.

We also investigated whether the potential immunogenicity of NAP can counteract the performance of CAR(NAP) T cells. Encouragingly, CAR(NAP) T-cell persistence was not affected, as CAR T cells were detected in the blood of approximately half of the tumour-free mice in both treatment groups (Supplementary Fig. 2). Importantly, the therapeutic efficacy of CAR(NAP) T cells was not diminished in mice with pre-existing anti-NAP antibodies

[1]Department of Immunology, Genetics and Pathology, Science for Life Laboratory, Uppsala University, Uppsala, Sweden. [2]These authors contributed equally: Chuan Jin, Jing Ma. These authors jointly supervised this work: Di Yu, Magnus Essand. ✉e-mail: di.yu@igp.uu.se; magnus.essand@igp.uu.se

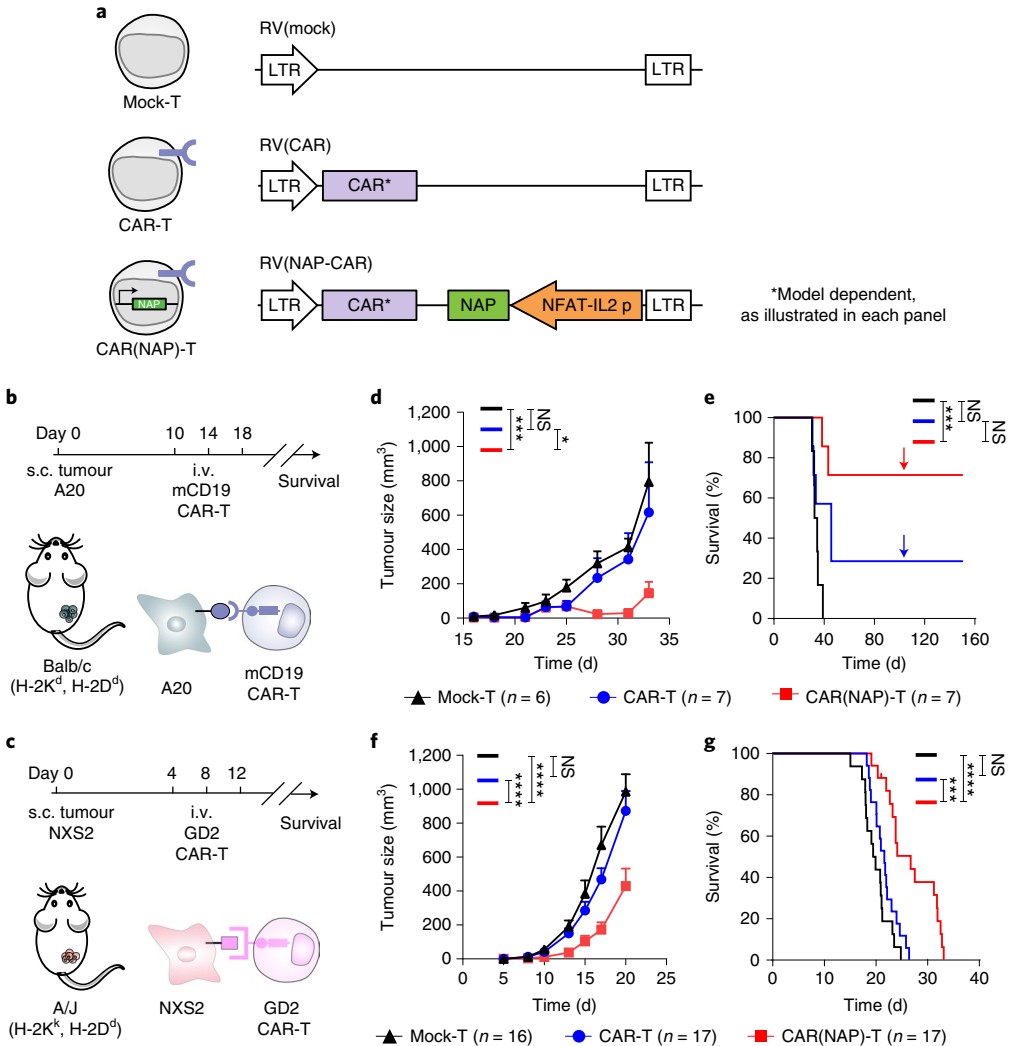

**Fig. 1 | Treatment with CAR(NAP) T cells targeting endogenous tumour-associated antigens delays tumour growth and prolongs survival in immunocompetent mouse cancer models. a**, Retroviral constructs used for mouse T-cell engineering. **b,c**, Treatment schedules for CAR-engineered T cells of subcutaneous murine A20 lymphoma that endogenously express murine CD19 (**b**), and murine NXS2 neuroblastoma that endogenously express GD2 (**c**). **d, e**, A20 tumour size (mean) until the first mouse had to be killed (**d**) and mouse survival (Kaplan–Meier curve) after treatment (**e**); arrows in **e** indicate rechallenge of cured mice with A20 tumour cells. **f, g**, NXS2 tumour size (mean) until the first mouse had to be killed (**f**) and mouse survival (Kaplan–Meier curve) after treatment (**g**). The A20 experiment was performed once, and the NXS2 experiment was performed twice and all data points were pooled. Tumour sizes between treatment groups were compared using two-way ANOVA, and survival curves were compared using the log-rank test. Error bars represent s.e.m. (NS, no statistical significance; *$P < 0.05$, ***$P < 0.001$, ****$P < 0.0001$). Precise $P$ values are reported in Supplementary Table 3.

(Supplementary Fig. 3). Additionally, repeated treatment with CAR(NAP) T cells did not result in elevated toxicity compared with conventional CAR T cells when assessing systemic cytokine release (Supplementary Fig. 4) or body weight (Supplementary Fig. 5). In conclusion, arming CAR T cells with NAP therefore constitutes a viable strategy for targeting various solid tumours.

Antigen heterogeneity is a major hurdle for CAR T-cell therapy of solid tumour[19]. To evaluate whether CAR(NAP) T cells can overcome antigen heterogeneity, we established a heterogeneous tumour model by subcutaneously injecting a mixture (1:1 ratio) of wild-type (antigen-negative) NXS2 cells and antigen-positive NXS2-mCD19 cells (Fig. 2a). CAR(NAP) T-cell treatment significantly reduced the growth of tumours with heterogeneous CAR-target antigen expression and prolonged mouse survival, while conventional CAR T-cell treatment failed to yield any therapeutic benefit compared with mock T-cell treatment (Fig. 2b,c and Supplementary Fig. 6). Furthermore, mice cured by CAR(NAP) T-cell treatment were

protected against rechallenge with wild-type NXS2 cells, indicating establishment of protective bystander immunological memory (Fig. 2c). Tumour-infiltrating CD8+ T cells isolated from CAR(NAP) T-cell-treated mice exhibited an antigen-experienced memory-like (CD62L+, CD44+) phenotype and were degranulated (CD107a+) to a higher degree than those isolated from CAR T-cell-treated mice (Fig. 2d–f). Importantly, splenocytes isolated from CAR(NAP) T-cell-treated mice, but not those from CAR T-cell-treated mice, responded to wild-type (CAR-targeted antigen-negative) NXS2 tumour cell lysate (Fig. 2g and Supplementary Fig. 7), indicating systemic activation of an endogenous bystander T-cell response, probably explaining why cured mice were protected when rechallenged with wild-type NXS2 (Fig. 2c).

A key feature of the bystander T-cell response is epitope spreading, that is, activation of endogenous CD8+ T cells directed against antigens other than the antigen targeted by the CAR, which is vital for eradication of antigen-heterogeneous solid tumours[19].

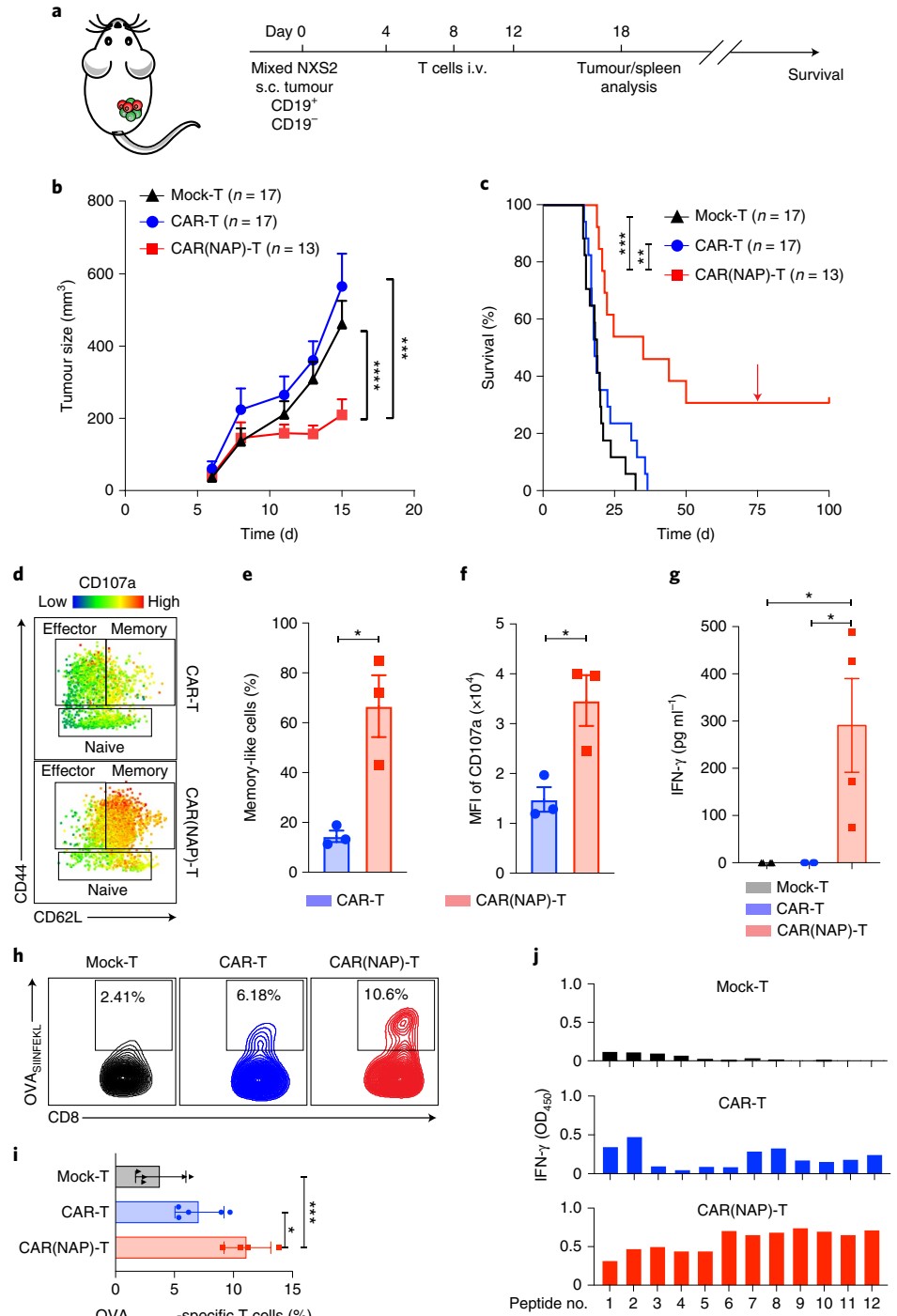

**Fig. 2 | CAR(NAP) T-cell treatment induces epitope spreading and bystander CD8+ T-cell killing of tumour cells lacking the antigen targeted by the CAR. a**, Treatment schedule of the mixed tumour model with 50% of tumour cells expressing the mCD19 CAR target antigen. **b,c**, Tumour size (mean) until the first mouse had to be killed (**b**) and mouse survival (Kaplan–Meier curve) after treatment and rechallenge (arrow) of cured mice with wild-type NXS2 tumour cells (**c**). The experiment was performed twice and all data were pooled. Tumour sizes were compared using two-way ANOVA, and survival curves were compared using log-rank test. **d–f**, Characteristics of tumour-infiltrating CD8+ T cells on day 18. Representative plots (**d**) and the percentage of memory-like (CD44+CD62L+) CD8+ T cells (**e**), and CD107a expression (**f**) on CD8+ T cells in treatment groups. Groups were compared using *t*-test. MFI, mean fluorescence intensity. **g**, Analysis of epitope spread as a consequence of treatment, with IFN-γ levels from co-cultures of splenocytes from treated mice and autologous DCs loaded with wild-type NXS2 (CD19−) tumour cell lysate. Values from co-cultures with unloaded DCs were seen as background and subtracted. Groups were compared using ANOVA with Bonferroni correction. **h,i**, Epitope spreading towards a non-targeted model antigen (OVA) for tumour-infiltrating CD8+ T cells after mCD19 CAR T-cell treatment of Panc02-mCD19-OVA tumours in C57BL/6 mice. Representative density plots (**h**) showing OVA(SIINFEKL)-specific CD8+ T cells and percentage (**i**) of these T cells in tumours from each group. Groups were compared using ANOVA with Bonferroni correction. **j**, IFN-γ production by splenocytes after mCD19 CAR T-cell treatment of NXS2-mCD19-OVA tumours in A/J mice upon rechallenge with different OVA peptides. In all panels, error bars represent s.e.m. (*$P < 0.05$, **$P < 0.01$, ***$P < 0.001$, ****$P < 0.0001$). Precise *P* values are reported in Supplementary Table 3.

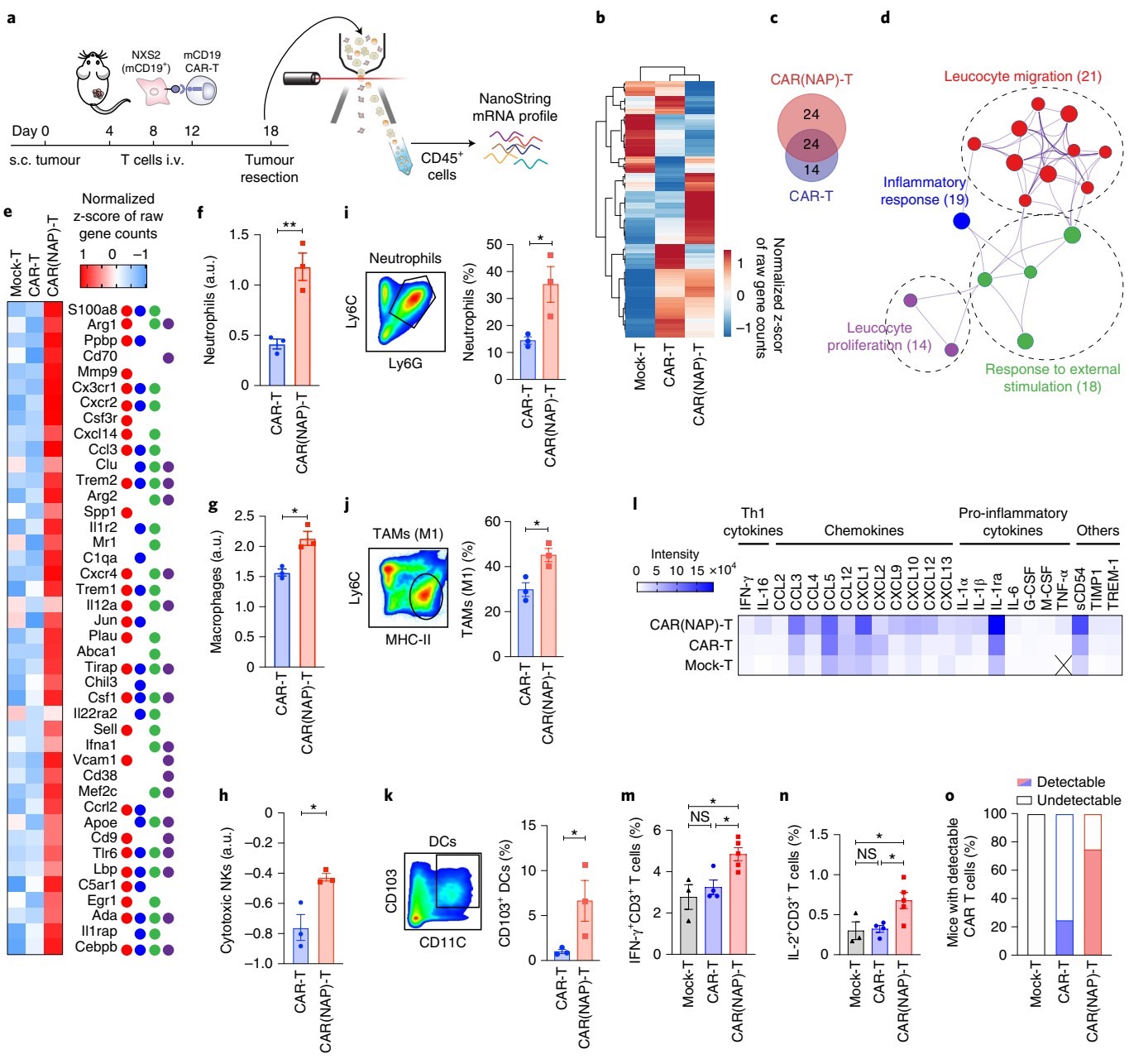

**Fig. 3 | CAR(NAP) T-cell treatment alters the immune composition of the tumour microenvironment leading to antitumour immunity. a–h,** CD45+ cells were isolated from NXS2-mCD19 tumours after CAR T-cell treatment and gene expression was analysed using NanoString. **a,** Schematic illustration of the experimental set-up. **b,** Heat map of gene expression presented as normalized z-score of raw gene counts. **c,** Upregulated genes (unique and overlapping) after treatment with CAR(NAP) T cells (red) and CAR T cells (blue) compared with mock T-cell treatment. **d,** Gene Ontology (GO) term enrichment network of uniquely upregulated genes after treatment with CAR(NAP) T cells versus CAR T cells, generated by Metascape analysis. Values in parentheses, $-\log_{10}(P)$. **e,** Heat map of gene expression presented as normalized z-score of raw gene counts and their GO term pathways classification, colour-coded as in **d. f–h,** Abundance of tumour-infiltrating neutrophils (**f**), macrophages (**g**) and cytotoxic NK cells (**h**), determined from NanoString data. a.u., arbitrary units. **i–k,** Flow cytometry analysis of tumour-infiltrating immune cells in the NXS2-mCD19 tumour model. Gating strategies (left panels) and percentage (right panels) of tumour-infiltrating Ly6C+Ly6G+ neutrophils (**i**), Ly6C^low^MHC-II+ M1 tumour-associated macrophages (**j**) and CD103+CD11C+ antigen-presenting DCs (**k**) are shown. **l,** Cytokine and chemokine levels in tumour lysate (X, below detection level). **m,** Percentage of IFN-γ producing cells in the CD3+ tumour-infiltrating T-cell population. **n,** Percentage of IL-2 producing cells in the CD3+ tumour-infiltrating T-cell population. **o,** Proportion of mice with qPCR-detectable tumour-infiltrating CAR-engineered T cells (n = 8). Groups were compared using t-test (*P < 0.05, **P < 0.01). In all panels, error bars represent s.e.m. Precise P values are reported in Supplementary Table 3.

We investigated epitope spreading using two tumour models: Panc02-mCD19-OVA and NXS2-mCD19-OVA, wherein ovalbumin (OVA) was introduced as a bystander antigen not targeted by the CAR T cells. Tumour infiltration of OVA-specific (SIINFEKL-directed) CD8+ T cells were significantly increased in mice treated with CAR(NAP) T cells compared with CAR

T cells (Fig. 2h,i). Furthermore, treatment with CAR(NAP) T cells induced a broad bystander CD8[+] T-cell response to all predicted H-2D[d]-restricted OVA-derived peptides, with higher interferon (IFN)-γ response compared with conventional CAR T-cell treatment (Fig. 2j), indicating epitope spreading resulting in both a stronger and broader bystander immune response. We identified six subpopulations of tumour-infiltrating CD8[+] T cells (Pop1–6, Extended Data Fig. 6a) on the basis of surface marker expression (Extended Data Fig. 6b,d). Memory-like (CD44[+], CD62L[+], CD127[+]), proliferating (Ki67[+]), activated and cytolytic (CD69[+], CD107a[+]) T cells (Pop1) were significantly enriched after CAR(NAP) T-cell treatment (Extended Data Fig. 6c). In contrast, exhausted (PD1[hi]Tim3[hi]LAG3[hi]) cytolytic T cells (Pop2) were enriched after conventional CAR T-cell treatment (Extended Data Fig. 6c,e). These observations support the notion that CAR(NAP) T-cell treatment induces epitope spreading, with an endogenous bystander response of highly activated CD8[+] T cells directed against various epitopes of tumour-associated antigens.

To study how CAR(NAP) T-cell treatment affects the tumour microenvironment, we investigated gene expression in tumour-infiltrating immune cells (CD45[+]) (Fig. 3a) using the NXS2-mCD19 model. Compared with mock T-cell treatment, both CAR(NAP) T cells and CAR T cells exhibited unique gene expression patterns (Fig. 3b,c). Gene set enrichment analysis for genes upregulated more than 2-fold in the CAR(NAP) T-cell treatment group compared with the CAR T-cell treatment group, revealed enrichment of Gene Ontology (GO) terms involved in leucocyte migration, proliferation, inflammatory response and regulation of response to an external stimulus (Fig. 3d,e). Cell type profiling from the gene expression analysis[20] indicated enhanced infiltration of neutrophils (Fig. 3f), macrophages (Fig. 3g) and cytotoxic natural killer (NK) cells (Fig. 3h) after CAR(NAP) T-cell treatment. Infiltration of neutrophils and M1 macrophages were also validated by flow cytometry (Fig. 3i,j). Flow cytometry analysis also revealed an increased infiltration of antigen-presenting DCs (CD103[+]CD11c[+]) after CAR(NAP) T-cell treatment (Fig. 3k). We also observed a similar immune cell infiltration profile in the A20 tumour model, where CAR(NAP) T-cell treatment led to higher infiltration of CD8[+] T cells and Gr1[+]MPO[+] neutrophils, and lower infiltration of FoxP3[+]CD4[+] regulatory T cells (Extended Data Fig. 7). Elevated infiltration of activated immune cells, in turn, resulted in higher levels of Th1-related cytokines and chemokines (Fig. 3l), and higher infiltration of IFN-γ (Fig. 3m) and IL-2 expressing (Fig. 3n) T cells in the tumour tissues after CAR(NAP) T-cell treatment compared with CAR T-cell treatment. Of note, CAR-positive T-cell infiltration in the tumours was detected in a higher proportion of mice after CAR(NAP) T-cell treatment compared with CAR T-cell treatment (Fig. 3o), consistent with an increased proliferative capacity and less exhausted phenotype. Collectively, these observations suggest that treatment with CAR(NAP) T cells induce a more pronounced immunologically 'hot', Th1-type pro-inflammatory tumour microenvironment compared with conventional CAR T cells, which explains the superior therapeutic efficacy of CAR(NAP) T cells.

Finally, to test the applicability of human CAR(NAP) T cells, we performed an in vitro proof-of-concept validation in freshly isolated human peripheral blood mononuclear cells (PBMCs) (Fig. 4a). NAP was expressed by human CAR(NAP) T cells upon target-antigen recognition (Fig. 4b,c), and human CAR(NAP) T cells and conventional CAR T cells killed human cancer cells equally well in vitro (Fig. 4d), even though transduction efficacy was higher for CAR T cells than for CAR(NAP) T cells (Supplementary Fig. 8a). Phenotypic characterization of CAR(NAP) T cells and CAR T cells upon transduction revealed similar CD4 and CD8 (Supplementary Fig. 8b), memory and effector T-cell composition (Supplementary Fig. 8c), and activation/exhaustion phenotype (Supplementary Fig. 8d). However, upon target cell recognition in vitro, gene expression analysis (Fig. 4e) revealed a distinct gene expression pattern for CAR(NAP) T cells compared with conventional CAR T cells, with GO terms enriched for adaptive immune response, cellular response to IFN-γ, regulation of cytokine production, type-I IFN signalling, NK cell-mediated immunity and granulocyte migration (Fig. 4f,g). In addition, the supernatants from co-cultures of CAR(NAP) T cells and tumour cells expressing the antigen targeted by the CAR were rich in potent chemo-attractants that induced migration and activation of DCs (Fig. 4h and Extended Data Fig. 8), neutrophils (Extended Data Fig. 9) and monocytes (Extended Data Fig. 10), confirming the findings of the gene set enrichment analysis (Fig. 4f,g).

In the presence of autologous DCs, CAR(NAP) T cells exhibited higher cytotoxicity than conventional CAR T cells (Fig. 4i), with enhanced IFN-γ secretion (Fig. 4j) and degranulation (Fig. 4k). Furthermore, gene expression analysis revealed that genes representing pathways involved in cytokine-mediated signalling were upregulated (Supplementary Fig. 9), and a Th1 transcriptional profile was apparent (Fig. 4l). The levels of IL-12 (Fig. 4m) and those of various pro-inflammatory cytokines and chemokines (Fig. 4n) were increased in supernatants from the co-culture of CAR(NAP) T cells, DCs and tumour cells, verifying the observed gene expression profile. Taken together, these observations indicate that the cytotoxic and immunostimulatory capacity of human CAR(NAP) T cells are superior to those of conventional human CAR T cells, which is consistent with our findings in the various mouse models.

The above observations align with published data that NAP, provided as a recombinant protein[21] or delivered as a therapeutic transgene[16,17], promotes DC maturation and Th1 polarization, and recruits and activates monocytes and neutrophils[18]. Accordingly, we propose that the enhanced therapeutic efficacy of CAR(NAP) T cells is associated with the ability of secreted NAP to attract and activate innate immune cells to kill tumour cells, and to activate DCs to facilitate the induction of epitope spreading, leading to CAR-target-independent CD8[+] T-cell bystander immunity (Fig. 5).

## Discussion
CAR T cells have been armed with several host-derived endogenous molecules (IL-12[12], IL-18[8–10], Flt3L[11], IL-7/CCL19[13], Heparinase[14] and CD40L[15]) that engage host immune cells to counteract antigen

**Fig. 4 | Human CAR T cells armed with NAP exhibit similar advantageous immunological characteristics as murine CAR(NAP) T cells. a**, Self-inactivating (ΔU3) lentiviral constructs used for human T-cell engineering. **b,c**, Representative histogram (**b**) and MFI (**c**) of NAP expression in CAR-positive T cells upon target cell recognition. Dashed line, MFI of isotype staining. **d**, Relative viability of target cells after exposure to CAR-engineered T cells. **e–g**, Gene expression in CAR T cells and CAR(NAP) T cells after exposure to target cells. **e**, Schamatic illustration of the experimental setting. **f**, GO term enrichment network of the uniquely upregulated genes in CAR(NAP) T cells according to Metascape analysis of the NanoString gene expression data. Numbers in brackets are −log₁₀($P$) values. **g**, Heat map of genes uniquely upregulated in CAR(NAP) T cells presented as normalized z-score of raw gene counts. **h**, Percentage of DCs migrating towards supernatant collected from a 5:1 co-culture of engineered T cells and Daudi target cells. **i–k**, Potency of CAR(NAP) T cells assisted by autologous DCs, presented as relative viability of target cells (**i**), IFN-γ secretion (**j**) and CD107a expression (**k**). **l–n**, CAR-engineered human T cells were co-cultured with autologous immature DCs and Daudi target cells (5:1:1) and assayed for T-cell polarization (**l**), and production of IL-12 (**m**) and other cytokines and chemokines (**n**). All experiments were performed three times and data were pooled. Error bars represent s.e.m. (*$P < 0.05$, **$P < 0.01$, ***$P < 0.001$, ****$P < 0.0001$). Precise $P$ values are reported in Supplementary Table 3.

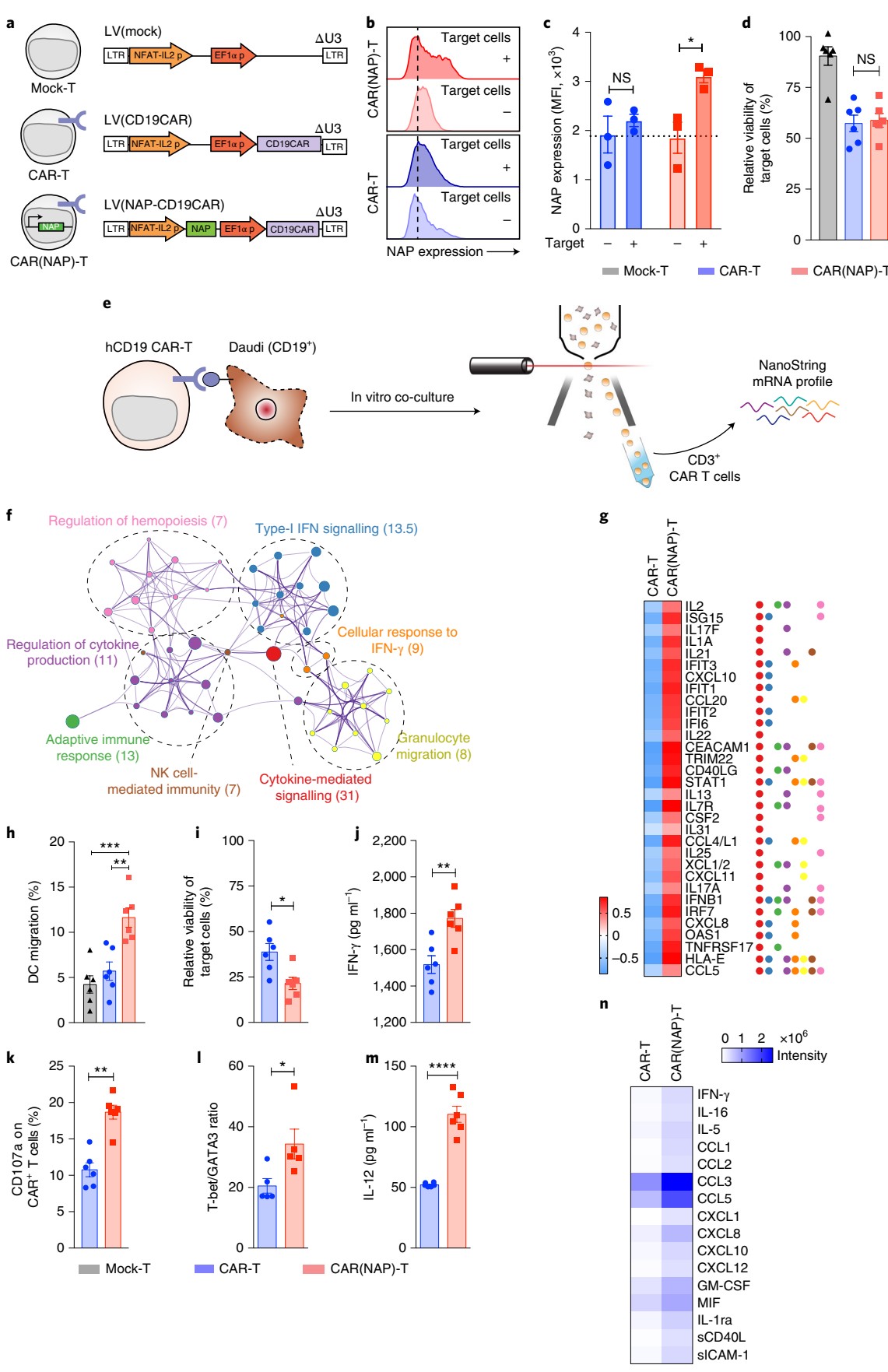

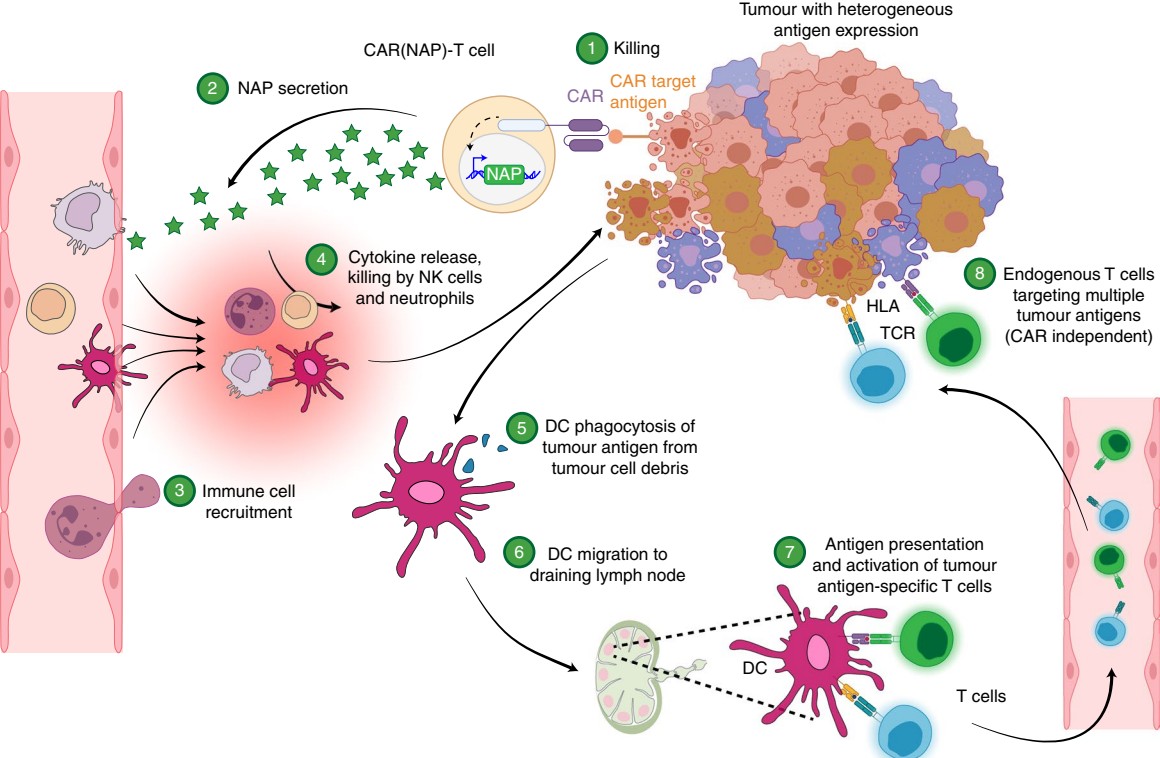

**Fig. 5 | Proposed mode of action of CAR(NAP) T-cell therapy.** CAR(NAP) T cells are CAR T cells armed with NAP, an immune activator derived from *H. pylori*. Within a solid tumour, heterogeneity in antigen expression makes it difficult to eradicate all tumour cells with conventional CAR T cells. CAR(NAP) T cells secrete NAP during killing of tumour cells (Steps 1 and 2). NAP works as a chemoattractant, recruiting innate immune cells, such as neutrophils and NK cells, and activates them to directly kill tumour cells independent of whether they express the CAR target or not (Steps 3 and 4). NAP also recruits monocytes and DCs, and activates them to secrete pro-inflammatory cytokines that directly counteract the hostile tumour microenvironment and strengthen the function of CAR(NAP) T cells. Immature DCs can phagocytose cell debris from dying tumour cells (Step 5). During this process, the DCs mature and migrate to a draining lymph node (Step 6) where they degrade the tumour cell debris and present peptides, including tumour-associated antigens, on the human leukocyte antigen (HLA) molecules to naive T cells. T cells with matching T-cell receptors (TCRs) recognize tumour-associated antigen epitopes and become activated (Step 7). The newly activated T cells recirculate back to the tumour site and kill the tumour cells through TCR-mediated recognition, which is independent of the antigen targeted by the CAR (Step 8). Therefore, this technology can counteract heterogeneity in antigen expression within solid tumours.

heterogeneity in solid tumours. These factors primarily work by engaging antigen-presenting cells[8–13]. Here we demonstrate that expression of a bacteria-derived virulence factor by CAR T cells can mediate bystander immunity with epitope spreading, leading to eradicating antigen-heterogeneous tumours. Our results suggest that NAP engages a broader range of immune effectors, for example, DCs, monocytes, NK cells and neutrophils to potentiate stronger antitumour immunity. NAP has been shown to substantially enhance the immunogenicity of poor immunogens[22]. This can be favourable as our data suggest that the immunogenicity of NAP can reverse the suppressive immune landscape in solid tumours. On the other hand, NAP immunogenicity can also be unfavourable, as immune response against NAP may hamper the persistence of the CAR(NAP) T cells, similar to what has been observed for T cells expressing other foreign proteins[23], or cause adverse effects such as anaphylaxis[24]. However, we show that persistence and efficacy of CAR(NAP) T cells were not altered in mice pre-immunized to have anti-NAP antibodies. It is even possible that pre-existing anti-NAP immunity can neutralize potential NAP-related systemic toxicity. However, the actual situation needs to be evaluated in a careful dose-escalation clinical study with CAR(NAP) T cells.

We used an inducible nuclear factor of activated T-cells (NFAT) promoter[25] to control NAP expression as proof of concept. The same promoter was used in a γ-retroviral vector to control IL-12 expres-

sion in a clinical trial evaluating IL-12-armed tumour-infiltrating lymphocytes[26]. Toxicities observed in patients in that trial were mainly attributed to basal level of systemic IL-12 expression[26]. Therefore, other tightly regulated promoters may be worth further investigation[27,28]. Whether NAP expressed from CAR(NAP) T cells would cause toxicity in a similar fashion as IL-12 in cancer patients is unknown. However, analysis of systemic cytokine levels in blood and body weight of mice during treatment did not show any CAR(NAP) T-cell-treatment-related toxicity. Furthermore, to the best of our knowledge, there are no reports describing NAP-related toxicity in patients with *H. pylori*-induced gastric inflammation, which further supports a safe profile of using NAP as a therapeutic transgene in CAR T cells.

Arming CAR T cells with a potent bacterial virulence factor is a unique concept in CAR T-cell design when compared with using host-derived factors[8–14]. There may be other pathogen-derived factors that are as effective, therefore our study opens up possibilities for evaluating intricate multifaceted pathogen-derived immune activators as ammunition for CAR T cells. We demonstrate that arming CAR T cell with NAP improves the treatment of solid tumours with heterogenous antigen expression. This approach may be used to engineer CAR T cells regardless of the CAR target antigen and tumour types, thus expanding translatability to various human cancers.

## Methods

**Antibodies.** Detailed information on the antibodies used in this study is provided in Supplementary Table 1.

**Primary cell isolation and cell culture.** The human B-cell lymphoblast cell line Daudi (ATCC CL-213), human lymphoma cell line BC-3 (ATCC CRL-2277) and murine lymphoma cell line A20 (ATCC TIB-208) were cultured in RPMI-1640 medium supplemented with 1 mM sodium pyruvate, 100 U ml⁻¹ penicillin, 100 µg l⁻¹ streptomycin (1% PeSt) and 10% (v/v) heat-inactivated foetal bovine serum (FBS). The mouse neuroblastoma cell line NXS2 (GD2⁺, a gift from Holger N. Lode, University of Greifswald), mouse pancreatic cancer cell line Panc02 (a gift from Rainer Heuchel, Karolinska Institute) and mouse colon carcinoma cell line CT26 (ATCC CRL-2638) were cultured in DMEM supplemented with 1 mM sodium pyruvate, 1% PeSt and 10% FBS.

To generate NXS2-mCD19 and Panc02-mCD19 cell lines stably expressing mouse *Cd19* (NM_009844.2), NXS2 cells and Panc02 cells were transduced with a lentiviral vector (LV) pBMN(mCD19-T2A-FLuc-F2A-puro) encoding murine *Cd19*, and selected on 5 µg ml⁻¹ puromycin. The expression of mouse CD19 was evaluated by flow cytometry (CytoFLEX LX, Beckman Coulter) using an anti-murine CD19 antibody. To generate CT26-hPSCA stably expressing human prostate stem cell antigen (hPSCA), CT26 cells were transduced with a lentiviral vector pBMN(hPSCA-T2A-FLuc-F2A-puro) encoding human PSCA, and selected on 10 µg ml⁻¹ puromycin. To generate ovalbumin (OVA)-expressing target cells, NXS2 and Panc02 cells were first transduced with a lentiviral vector pBMN(OVA-T2A-FLuc-F2A-puro) encoding full length chicken ovalbumin (NM_205152.2), and selected on 5 µg ml⁻¹ puromycin. The cells were then transduced with pBMN(mCD19-T2A-FLuc-F2A-puro) as mentioned above and CD19-positive cells were sorted by flow cytometry (BD FACS Melody, BD Biosciences). All cell culture reagents were from Thermo Fisher, unless stated otherwise.

PBMCs were isolated using Ficoll-Paque (GE Healthcare) from fresh buffy coats of healthy anonymized donors at the Blood Centre at Uppsala University Hospital. PBMCs were cultured in RPMI-1640 supplemented with 10% FBS, 1 mM sodium pyruvate and 1% PeSt. CD14⁺ cells in the PBMC pool were isolated using specific beads (Miltenyi Biotec) and differentiated over 5 d into immature DCs (imDCs) in the presence of 50 ng ml⁻¹ granulocyte-macrophage colony-stimulating factor (Gentaur) and 25 ng ml⁻¹ IL-4 (Gentaur). Fresh medium with cytokines was replaced every 2 d. All cells were cultured in a humidified incubator under a 5% CO₂ atmosphere at 37 °C.

**Vector design, retroviral and lentiviral production.** The murine CD19-targeting CAR sequence, containing murine CD3z and CD28 signalling domains, has been described previously[29]. The *nap* transgene sequence was placed under the control of an inducible NFAT-IL-2 promoter, designed as described previously[30]. The NFAT-IL-2 promoter consists of six repeats of the NFAT element and the minimal murine IL-2 promoter[31]. The orientation of the *nap* cassette was opposite to that of the murine CD19 CAR sequence. Murine stem cell virus-based retroviral vector, pMIG-W (Luk Parijs, Addgene, plasmid 12282)[32] was used as retroviral vector backbone to construct RV(CD19CAR) and RV(NAP-CD19CAR). RV(mock) is an empty retroviral vector used for generating mock T cells.

The GD2-targeting single chain fragment (scFv) sequence (derived from antibody clone 14G2a) was kindly provided by Dr Eric Yvon and Dr Malcolm Brenner from Baylor College of Medicine (Houston, TX). The human PSCA CAR scFv sequence was described previously[33]. The CD19 scFvs in RV(CD19CAR) or RV(NAP-CD19CAR) constructs were replaced with corresponding scFvs to generate RV(GD2CAR), RV(NAP-GD2CAR), RV(hPSCACAR) and RV(NAP-hPSCACAR). All sequences were synthesized and subcloned by Genscript. Retroviruses were produced using a packaging plasmid pCL-Eco (Inder Verma, Addgene plasmid 12371)[34] and Gryphon retroviral packaging cell line (Allele Biotechnology) following the manufacturers' instructions.

The human CD19-targeting CAR sequence, which contains human CD3z and 4-1BB signalling domains, was designed as described previously[35]. The encoding sequence was cloned into a third-generation self-inactivating (SIN) lentiviral vector under the control of the elongation factor-1 alpha (EF1a) promoter to generate LV(CD19CAR). The *nap* transgene sequence was codon-optimized for expression in human cells, and placed under the control of the inducible NFAT-IL-2 promoter. The promoter and the *nap* sequence were cloned into LV(CD19CAR) to generate LV(NAP-CD19CAR). LV(mock) is an empty lentiviral vector used for generating mock T cells. All sequences were synthesized by Genscript. The generation of lentiviral particles has been described previously[36].

Unique materials are available under a material transfer agreement with Elicera Therapeutics.

**Nano-luciferase expression by CAR(nLuc) T cells in vitro and in vivo.** The *nap* gene in LV(NAP-CD19CAR) was replaced with the gene encoding NanoLuc luciferase to construct LV(nLuc-CD19CAR). For in vitro experiments, NanoLuc luciferase expression was measured in CAR(nLuc) T cells alone or CAR(nLuc) T cells co-cultured with Daudi target cells at a ratio of 5:1 for 24 h. The nLuc was detected using Nano-Glo luciferase assay kit (Promega).

For in vivo experiments, female (5–6 weeks old) NOD-SCID mice (Janvier Labs) were injected intravenously (i.v.) into the lateral tail vein with 5×10⁶ CAR(nLuc) T cells together with Daudi (CD19⁺) target cells or BC-3 (CD19⁻) target cells (n=3) mixed at a ratio of 2:1 in a total volume of 100 µl on day 0. NanoLuc luciferase expression was measured by i.v. injection of 5 µl of substrate (Promega) in 100 µl PBS solution into the lateral tail vein on day 3 and imaged using the NightOWL in vivo imaging system (Berthold Technologies).

**NAP expression in murine CAR T cells.** T cells were mixed with A20 cells (CD19⁺) at a ratio of 5:1 and cultured for 24 h. NAP secretion was then blocked by 12 h incubation with Brefeldin A (BD GolgiPlug, BD BioSciences) following the manufacturer's instructions. The cells were then stained for NAP expression using a primary α-NAP antibody (a gift from Dr Marina de Bernard, University of Padova, Italy, 1:10) and a secondary monkey anti-rabbit IgG antibody (Thermo Fisher, 1:500). The T cells were identified by staining with CD3 antibody (BioLegend). Stained cells were analysed in a CytoFLEX S flow cytometer (Beckman Coulter) and data were analysed using FlowJo v10.2-v10.8.1.

**Adoptive transfer of murine CAR T cells in murine tumour models.** Spleens from 6–8-week-old female Balb/c mice, C57Bl/6NRj mice (Taconic) or A/J mice (Envigo) were collected, mashed over a 70 µm cell strainer, and resuspended in complete RPMI after red blood cell lysis using ACK lysis buffer (Thermo Fisher). Splenocytes (3×10⁶ per ml) were activated with concanavalin A (2 µg ml⁻¹) (Sigma-Aldrich) and murine IL-7 (1 µg ml⁻¹) (Miltenyi Biotec). Three days later, the activated splenocytes were collected, transduced with retrovirus as described[37], and administered to mice i.v. after 18–24 h in a volume of 100 µl DPBS suspension.

*A20 tumour model.* Female Balb/c mice (6–8 weeks old) were inoculated subcutaneously (s.c.) with 2×10⁵ A20 tumour cells in 100 µl DPBS into the right hind flank. The mice were then treated with 3×10⁶ T cells injected into the lateral tail vein 10, 14 and 18 d after tumour cell implantation. For survival analysis, the mice were monitored regularly and euthanized upon reaching the humane endpoint (mock T cells: n=6, CAR T cells and CAR(NAP) T cells: n=7). For tumour microenvironment analysis, tumour tissues were collected for analysis 3–5 d after the last treatment (n=3).

*NXS2-GD2 model.* Female A/J mice (6–8 weeks old) were inoculated s.c. with 1×10⁶ NXS2 cells in 100 µl DPBS into the right hind flank. The mice were then treated with 3×10⁶ T cells injected into the lateral tail vein 4, 8 and 12 d after tumour cell implantation. The mice were monitored regularly and euthanized upon reaching the humane endpoint. The experiment was repeated twice and all data were pooled together (mock T cells: n=16, CAR T cells and CAR(NAP) T cells: n=17).

*CT26-hPSCA model.* Female Balb/c mice (6–8 weeks old) were inoculated s.c. with 5×10⁵ CT26-hPSCA cells in 100 µl DPBS into the right hind flank. The mice were then treated with 3×10⁶ T cells injected into the lateral tail vein 10, 14 and 18 d after tumour cell implantation. The mice were monitored regularly and euthanized upon reaching the humane endpoint. The experiment was repeated twice and all data were pooled together (mock T cells: n=17, CAR T cells: n=15, CAR(NAP) T cells: n=16).

*NXS2-mCD19 model.* Female A/J mice (6–8 weeks old) were inoculated s.c. with 1×10⁶ NXS2-mCD19 cells in 100 µl DPBS into the right hind flank. The mice were then treated with 3×10⁶ T cells injected into the lateral tail vein 4, 8 and 12 d after tumour cell implantation. For survival analysis, the mice were monitored regularly and euthanized upon reaching the humane endpoint. The experiment was repeated twice and all data were pooled together (mock T cells: n=6, CAR T cells: n=14, CAR(NAP) T cells: n=12). For tumour microenvironment analysis, tumour tissues were collected on day 18 for analysis (n=3).

*Panc02-mCD19 model.* Female C57Bl/6NRj mice (6–8 weeks old) were inoculated s.c. with 2×10⁵ Panc02-mCD19 cells in 100 µl DPBS into the right hind flank. The mice were then treated with 3×10⁶ T cells injected into the lateral tail vein 4, 8 and 12 d after tumour cell implantation. The mice were monitored regularly and euthanized upon reaching the humane endpoint. The experiment was repeated twice and all data were pooled together (mock T cells: n=14, CAR T cells: n=8, CAR(NAP) T cells: n=12).

*Panc02-mCD19-OVA model.* Female C57Bl/6NRj mice (6–8 weeks old) were inoculated s.c. with 2×10⁵ Panc02-mCD19-OVA cells in 100 µl DPBS into the right hind flank. The mice were then treated with 3×10⁶ T cells injected into the lateral tail vein 4, 8 and 12 d after tumour cell implantation. The mice were euthanized on day 16 to collect tumour tissues and spleens for analysis (n=4).

*NXS2-mCD19-OVA model.* Female A/J mice (6–8 weeks old) were inoculated s.c. into the right hind flank with 1×10⁶ NXS2-mCD19-OVA cells in 100 µl DPBS. The mice were then treated with 3×10⁶ T cells injected into the lateral tail vein 4, 8 and 12 d after tumour cell implantation. The mice were euthanized on day 19 to collect tumour tissues and spleens for epitope spreading analysis (n=3).

*NXS2 and NXS2-mCD19 mixed tumour model.* Female A/J mice (6–8 weeks old) were inoculated s.c. with $1 \times 10^6$ tumour cells (1:1 mixture of NXS2 and NXS2-mCD19 cells) in 100 μl DPBS into the right hind flank. The mice were then treated with $3 \times 10^6$ T cells injected into the lateral tail vein 4, 8 and 12 d after tumour cell implantation. For survival analysis, the mice were monitored regularly and euthanized upon reaching the humane endpoint. The experiment was repeated twice and all data were pooled together (mock T cells: $n = 17$, CAR T cells: $n = 17$, CAR(NAP) T cells: $n = 13$). For tumour microenvironment analysis, tumour tissues were collected for analysis 7 d after the last treatment ($n = 3$). For recall assay, spleens were collected for analysis 7 d after the last treatment ($n = 4$).

*Survival analysis.* The animals were monitored individually for tumour growth and body weight until humane endpoints were reached or until the tumour volume exceeded the study endpoint volume (EPV, 1,000 mm³); tumour size was calculated as volume = length × width² × π/6. The time to endpoint (TTE) for each mouse was calculated as TTE = (log(EPV)−b)/m, where the constant b is the intercept and m is the slope of the line obtained by linear regression (time vs tumour volume) of a log-transformed tumour growth data set, which comprised the first measured tumour volume when EPV was exceeded and three consecutive measured tumour volumes immediately before the attainment of EPV. Any animal determined to have died from treatment-related causes was assigned a TTE value equal to the day of death. Any animal that died from non-treatment-related causes was excluded from the analysis. Survival curve was generated on the basis of the TTE value using the Kaplan–Meier method, and compared using the log-rank (Mantel–Cox) test.

*Rechallenge model.* Cured mice after treatment in the A20 model and the NXS2/NXS2-mCD19 mixed model, and newly purchased naïve mice (as control) were inoculated s.c. with corresponding tumour cells ($2 \times 10^5$ A20 or $1 \times 10^6$ NXS2) in 100 μl DPBS into the left hind flank. Tumour growth was monitored regularly until tumour size in the control group reached the humane endpoint (1,000 mm³).

**Mouse T-cell cytotoxicity assay and IFN-γ ELISA.** The cytotoxicity assay was performed using luciferase-expressing target cells. Mouse splenocytes were collected, activated and transduced with CAR-encoding retroviral vectors as described above. The CAR-engineered T cells were co-cultured with firefly luciferase-tagged target cells at indicated ratios for 5 d, in a total volume of 200 μl in a 96-well plate. Luciferase expression and activity (as an indicator of target cell viability) were determined using ONE-Glo reagent (Promega) as previously described[38]. Specific lysis of each sample was calculated using the luminescence of co-cultured samples against target cells alone. The supernatants were collected after 18 h of co-culture, and IFN-γ levels determined by an ELISA kit (Mabtech).

**Analysis of CD45± tumour-infiltrating cells.** *NanoString.* Tumour samples were collected from NXS2-mCD19 animals on day 18 of the experiment. The samples were collected and enzymatically digested (Liberase, Roche) into single-cell suspensions. CD45+ cells were sorted using BD FACS Melody (BD Biosciences) and RNA was isolated from these cells (RNeasy Plus mini kit, Qiagen). Then, mRNA levels were directly measured using the Mouse-Pan cancer immune-oncology kit from NanoString nCounter gene expression system (NanoString). Differential expression analyses of mRNA were performed using nSolver analysis software (NanoString) and visualized by ClustVis[39]. Gene Ontology (GO) annotation analysis of the target genes was performed using the Metascape tool (http://metascape.org), which facilitates enrichment analysis of biological processes and pathways of input genes[40]. Only genes upregulated more than 2 times in the CAR(NAP) T-cell group compared with the CAR T-cell group were imported into Metascape and output P value cut-off was set to P < 0.0001. Cell type profiling analyses were performed using nSolver analysis software (NanoString).

*Tumour-infiltrating CAR T cells and CAR(NAP) T cells.* RNA isolated as mentioned above was analysed using quantitative PCR (qPCR) for detection of the presence of engineered CAR T cells, with primers specifically targeting the CAR molecule: pF: 5'-CTACATGAACATGACCCCCAGAAGGCC-3' and pR: 5'-CCATCTTGTCTTTCTGCAGGGCGTTGTAG-3'. GAPDH gene expression (detected with primers: pF 5'-ACCACAGTCCATGCCATCAC-3' and pR: 5'-TCCACCACCCTGTTGCTGTA-3') was used as internal control. The proportions of mice with detectable signals are reported.

**Flow cytometry analyses of the tumour microenvironment.** For the Panc02-mCD19-OVA animal experiment, mice were killed on day 16. Tumours were collected and enzymatically digested (Liberase, Roche) into single-cell suspensions. CD45+ cells were bead-isolated (Miltenyi Biotec) and stained for CD45, CD3e, CD8, CD4 and OVA(SIINFEKL)-tetramer (1:100) (murine OVA T panel in Supplementary Table 1). OVA-specific CD8+ T cells were gated on CD3+ T cells.

To evaluate the tumour microenvironment, mice were killed, and tumours were collected and enzymatically digested (Liberase, Roche) into single-cell suspensions following the manufacturer's instructions. CD45+ cells were isolated using magnetic beads (Miltenyi Biotec) and stained for CD45, CD3, CD8, CD4, CD44, CD62L and CD107a (mouse T-cell panel 1 in Supplementary Table 1) to evaluate tumour-infiltrating T cells; and stained for CD45, CD11c, CD11b, B220, Ly6C, Ly6G, MHCII and CD103 (mouse immune cells panel 1 in Supplementary Table 1) to evaluate tumour-infiltrating neutrophils, DCs and macrophages. Stained cells were analysed using BD FACSCanto II (BD Biosciences).

For the NXS2-mCD19-OVA animal experiment, mice were killed on day 19. Tumours were collected and enzymatically digested (Liberase, Roche) into single-cell suspensions. CD45+ cells were bead-isolated (Miltenyi Biotec) and stained for CD45, CD3e, CD8, CD4, CD44, CD62L, CD69, CD127, PD1, TIM3, LAG3, Ki67, KLRG1, CD107a and Fixable dye 700 to discriminate live and dead cells (mouse T-cell panel 2 in Supplementary Table 1). Stained cells were analysed using BD LSRFortessa (BD Biosciences). t-distributed stochastic neighbor embedding (t-SNE) dimension reduction was performed using Cytosplore[41,42]. Cells were manually gated for CD8+, and then t-SNE was performed with perplexity = 30 for the following markers: CD44, CD62L, CD69, CD127, PD1, Tim3, LAG3, Ki67, KLRG1 and CD107a. The percentages of cells with single, double and triple positivity of PD1, TIM3 and LAG3 are presented as a pie chart generated using SPICE[43].

**Epitope spreading assay.** H-2Dᵈ–restricted OVA-derived peptides were predicted using the Immune Epitope Database (IEDB) tool. The top 12 peptides (Supplementary Table 2) with highest prediction scores were synthesized (JPT Peptide Technologies). Splenocytes were collected from NXS2-mCD19-OVA animals on day 19 of the experiment. Approximately $1 \times 10^5$ cells were restimulated with each peptide in triplicate, at 10 μg ml⁻¹ in a 96-well plate for 3 d. The supernatant was collected and analysed by mouse IFN-γ ELISA (Mabtech).

**Recall assay.** Bone marrow-derived DCs were generated from wild-type A/J mice and pulsed with NXS2 cell lysate (prepared by freeze-thawing) for 24 h at a ratio of 200 μg protein per $1 \times 10^6$ DCs. Splenocytes were collected from mice with NXS2 and NXS2-mCD19 mixed tumour after treatment. These splenocytes ($1.25 \times 10^5$) were co-cultured with the above-mentioned pulsed DCs ($2.5 \times 10^4$) for an additional 5 d. The co-culture supernatant was then assayed by IFN-γ ELISA (Mabtech).

**Immunohistochemistry analyses of the tumour microenvironment.** Frozen tumour tissues were collected from the A20 model after treatment and sliced into 6 μm sections. The sections were fixed with ice-cold acetone (Sigma-Aldrich) for 15 min and dried. The slides were dehydrated with PBS for 3 min (repeated 3 times), followed by blocking with 3% BSA-PBS for 2 h at r.t. The sections were stained with antibodies listed in the murine tumour immune cells staining panel (Supplementary Table 1): CD8 (1:200), GR1 (1:200) and MPO (1:100), CD4 (1:50) and FoxP3 (1:50), at 4 °C overnight, values in parentheses are antibody dilution factors in the final working solution. The sections were washed twice with PBS-T and incubated with Alexa-488-streptavidin (1:1,000) for 30 min to visualize the biotin-conjugated MPO antibody. The sections were then washed twice with PBS-T, stained with Hoechst 33342 (Thermo Fisher) for 15 min at r.t. and mounted with Fluromount-G (Southern Biotech). The sections were imaged in a Zeiss AxioImager microscope (Zeiss).

**Human T-cell transduction and expansion.** Human PBMCs ($5 \times 10^6$) isolated as described above were stimulated for 3 d with OKT-3 (100 ng ml⁻¹, BioLegend) in a culture medium containing IL-2 (100 IU ml⁻¹, Proleukin, Novartis). Activated T cells ($1 \times 10^6$) were resuspended in $2 \times 10^7$ IU concentrated lentivirus particles, together with 10 mg ml⁻¹ protamine sulphate (Sigma-Aldrich) and IL-2 (100 IU), for a total volume of 20 μl, and incubated for 4 h. Then, 1 ml of fresh culture medium was added. The T cells were transduced again the following day in the same manner, and then cultured in 1 ml of culture medium containing IL-2 (100 IU ml⁻¹) for 7 d. Before further experiments, T cells were expanded for 2 weeks using a rapid expansion protocol as previously described[38].

**Human T-cell cytotoxicity assay.** The cytotoxicity assay was performed using luciferase-expressing target cells. CAR-engineered human T cells were allowed to rest for 3 d after rapid expansion in a medium containing a low dose of IL-2 (20 IU ml⁻¹) before being used for functional assays. The T cells were co-cultured with firefly luciferase-tagged target cells (Daudi, CD19+; at a 5:1 ratio), or with Daudi and autologous imDCs (at a 5:1:1 ratio) for 4 d, in a total volume of 200 μl in a round-bottom 96-well plate. Luciferase expression and activity (as an indicator of target cell viability) were determined using ONE-Glo reagent (Promega) as previously described[38]. Specific lysis of each sample was calculated using the luminescence of co-cultured samples against target cells alone.

**NAP expression in human CAR(NAP) T cells.** NAP secretion by CAR(NAP) T cells was assessed after co-culture with Daudi cells (CD19+). The cells were mixed at a ratio of 5:1 and cultured for 24 h. NAP secretion was then blocked by 12 h incubation with Brefeldin A (BD Biosciences). The cells were then stained for NAP expression using antibodies in the human CAR T-cell panel (Supplementary Table 1): CD3, primary α-NAP antibody (1:10) and a secondary monkey anti-rabbit IgG antibody (1:500), rabbit anti-mouse Ig(H+L) Fab fragment (CAR expression). Stained cells were analysed using a CytoFLEX S flow cytometer (Beckman Coulter) and data were analysed using FlowJo v10.2-v10.8.1.

**CAR-engineered T-cell characterization.** The CAR expression level was detected using F(ab')$_2$ fragment goat anti-mouse IgG (Jackson ImmunoResearch). Cells were also stained for surface markers CD3, CD4 and CD8 using antibodies listed in CAR expression in the human CAR T-cell panel (Supplementary Table 1). CAR-positive T cells were gated on CD3$^+$ T-cell population. CD8 and CD4 T-cell compositions were gated on the CAR$^+$CD3$^+$ population.

LV-engineered human T cells were stained for CD3, CD4, CD8, CAR, CD45RA and CD62L at 7 d after viral transduction to evaluate memory/effector phenotype using antibodies listed in the human T-cell memory/effector phenotype panel (Supplementary Table 1). In yet another setting, these cells were stained for CD3, CD4, CD8, CAR, PD1 and TIM3 to evaluate exhaustion phenotype using antibodies listed in the exhaustion phenotypic human T-cell panel (Supplementary Table 1).

**Phenotypic analysis of immune cells in vitro.** Phenotypic analysis of human DCs after exposure to the supernatant from a co-culture of engineered T cells and target cells was performed using antibodies staining: CD14, CD1a, CD83, CD80, CD86 and CD70 (human DC markers panel in Supplementary Table 1). Engineered human T cells were co-cultured with Daudi and autologous imDCs (at a 5:1:1 ratio) for 20–24 h. The Th1/Th2 phenotype of T cells was analysed by staining of T cells with the surface markers: CD3, CD4 and CD8 (human T-cell identification panel in Supplementary Table 1), followed by 1 h cell permeabilization with True-Nuclear transcription factor buffer set (BioLegend), and staining with antibodies for T-bet and GATA3. The T-bet/GATA3 ratio was then calculated for the CD3$^+$CD4$^+$ T-cell population.

**Cytokine secretion and CD107a degranulation.** Engineered human T cells were co-cultured with Daudi target cells (at a 5:1 ratio), or with Daudi and autologous imDCs (at a 5:1:1 ratio) for 20–24 h. The supernatant was collected, and IFN-γ or IL-12 levels were analysed by ELISA (Mabtech). CAR T cells were stained with antibodies for: CD3, goat anti-mouse Ig(H+L) Fab fragment (CAR expression) and CD107a (human T-cell degranulation panel in Supplementary Table 1) before evaluation by flow cytometry (BD Canto II, BD Biosciences). The CD107a$^+$ T-cell population was then calculated as a percentage of the CAR$^+$ T-cell population.

**Cytokine profiling.** Tumours were collected from NXS2-mCD19 animals on day 19 after tumour implantation, and frozen on dry ice. The tumours were then treated with 500 µl RIPA buffer (Thermo Fisher) and sonicated twice at 40% amplitude for 30 s, with 30 s intervals. The tumour samples were rotated (end over end) for 30 min at 4 °C, and centrifuged at 1,500 r.p.m. for 10 min. The supernatants were collected and analysed using Proteome Profiler mouse cytokine array kit (R&D Systems).

Supernatant from 4 d co-cultures of the human CAR T cells with Daudi target cells and autologous imDCs (mixed at a 5:1:1 ratio) was collected, and the cytokine and chemokine levels were analysed using the Proteome Profiler human cytokine array kit (R&D Systems).

**Migration assay.** Cell migration was assessed using a 96-well ChemoTx disposable chemotaxis system (Neuro Probe) with a polycarbonate membrane filter (3 µm pore size). Supernatant from 4 d co-cultures of engineered T cells with Daudi target cells (mixed at a 5:1 ratio) was placed in the bottom wells and $2 \times 10^5$ freshly isolated neutrophils, CD14$^+$ monocytes (Milteny Biotec) or imDCs differentiated from human monocytes were seeded at the top of the transwell chamber. Cell migration was assessed after 1.5 h incubation at 37 °C, and was quantified using CellTiter-Glo luminescent cell viability assay (Promega). The percent migration of cells was calculated as $(RLU_{T\ cells+Daudi} - RLU_{only\ T\ cells})/(RLU_{total\ migration\ cells}) \times 100\%$, where RLU refers to relative luciferase units.

**Neutrophil activation assay.** Engineered T cells were co-cultured with Daudi target cells and freshly isolated autologous neutrophils at a 5:1:1 ratio for 48 h. The neutrophils were gated as CD15$^+$ cells and activation markers were analysed using antibodies: CD11b, HLA-DR, CD66b and CD54 (human neutrophils activation panel in Supplementary Table 1). The levels of IL-12, IL-1β and IFN-γ in co-cultures were analysed by ELISA (Mabtech) following the manufacturer's guidelines.

**Monocyte differentiation and activation assay.** Transwell culture (Corning Transwell cell culture inserts, 0.4 µm pore size) was prepared by placing the monocytes (CD14$^+$) in a 24-well plate, with the engineered T cells mixed with Daudi target cells at a ratio of 5:1 (T cell:Daudi) in the inserts permissible only to the supernatant. The monocyte phenotype was determined after 5 d by staining for CD14, CD1a, CD80, CD86, CD40 and CD70 (human DC markers panel in Supplementary Table 1).

**Analysis of mRNA levels in human CAR T cells.** The human engineered T cells with Daudi target cells and autologous imDCs (mixed at a 5:1:1 ratio) were stained with BV510-CD3 after 2 d co-culture. The CD3$^+$ T cells were sorted using BD FACS Melody (BD Bioscience). The total RNA was isolated from sorted cells using RNeasy mini kit (Qiagen). Then, mRNA levels in samples were directly

measured using the CAR T-cell characterization panel from NanoString nCounter gene expression system (NanoString). Analysis (except that the threshold was set to >4-fold change) and visualization were performed as with murine samples mentioned above.

**Efficacy assessment in animal model with pre-existing anti-NAP antibodies.**
*Animal model.* Female Balb/c mice (6–8 weeks old) were injected with $2 \times 10^{10}$ evg (encapsulated viral genome) with an empty adenoviral vector (Ad(mock)) or an adenoviral vector encoding NAP (Ad(NAP)) on day -14 and day -7 (relative to tumour implantation on day 0). Both viral vectors were delivered half-dose intraperitoneally and half-dose subcutaneously. The adenoviral vectors were constructed and produced as previously reported[17]. Blood was collected on day −1 for analysis of the presence of anti-NAP antibodies.

*Anti-NAP antibody ELISA:* 96-well plates (ELISA high binding capacity) were coated with 0.3 µg purified NAP antigen (ProSpec) in 100 µl carbonate-bicarbonate buffer at r.t. overnight. The plate was washed with 0.05% PBS-T and blocked with 1% BSA-PBS for 2 h. Diluted mouse serum samples were added and incubated at r.t. for 1 h, followed by triple washing with PBS-T. Secondary anti-mouse IgG-HRP (1:2,000) (Invitrogen) was added, and the plates were incubated at r.t. for 1 h. The reaction was developed using TMB peroxidase substrate (Invitrogen) for 15 min and stopped by 0.16 M sulfuric acid. Absorbance was measured at 450 nm.

*Survival experiment.* On day 0, A20 tumour cells ($2 \times 10^5$ cells in 100 µl DPBS) were injected into the right hind flank. The mice were treated with $3 \times 10^6$ T cells injected into the lateral tail vein on days 10 and 14. For survival analysis, the mice were monitored regularly and euthanized upon reaching the humane endpoint. The experiments were repeated 2 times and data were pooled for analysis (CAR T cells: $n = 14$ and CAR(NAP) T cells: $n = 13$).

**Long-term persistence analysis of CAR T cells and CAR(NAP) T cells.** Blood samples were collected on day 70 from tumour-free mice in the A20 model (CAR T cells: $n = 6$ and CAR(NAP) T cells: $n = 8$), and on day 60 from mice immunized to have pre-existing anti-NAP antibodies (pre-immunized: $n = 4$ and non-immunized: $n = 5$). RNA was isolated as mentioned above and analysed using qPCR to detect the presence of CAR T cells, with primers specifically targeting the CAR molecule: pF: 5'-CTACATGAACATGACCCCCAGAAGGCC -3' and pR: 5'-CCATCTTGTCTTTCTGCAGGGCGTTGTAG-3'. GAPDH gene expression (detected with primers: pF 5'-ACCACAGTCCATGCCATCAC-3' and pR: 5'-TCCACCACCCTGTTGCTGTA-3') was used as internal control. The proportions of mice with detectable signals are reported.

**Detection of cytokines in the blood.** Female Balb/c mice (6–8 weeks old) were inoculated s.c. with $2 \times 10^5$ A20 tumour cells in 100 µl DPBS into the right hind flank. The mice were treated with $3 \times 10^6$ T cells injected into the lateral tail vein on days 10, 14 and 18 after tumour cell implantation. Blood was collected on day 30, and the serum was analysed using the LEGENDplex mouse cytokine storm panel (13-plex) (BioLegend). The data were analysed using LEGENDplex data analysis software (CAR T cells and CAR(NAP) T cells: $n = 5$).

**Ethical approval.** The Uppsala Research Animal Ethics Committee approved all animal studies (N164/15; N185/16; 5.8.18-19434/2019). The human buffy coats obtained from healthy blood donors were anonymized.

**Statistical analyses.** The data are reported as mean ± s.e.m. Statistical analysis was performed using GraphPad Prism software version 6.07-9.3.1. Statistical analyses in all figures, unless otherwise specified in Supplementary Table 3, were performed using multiple comparison of parametric analysis of variance (ANOVA) (more than two data groups) with recommended post-hoc corrections, or non-parametric $t$-tests (only two data groups compared). Values of $P < 0.05$ were considered to be statistically significant and precise $P$ values are reported in Supplementary Table 3.

**Reporting Summary.** Further information on research design is available in the Nature Research Reporting Summary linked to this article.

## Data availability
The main data supporting the results in this study are available within the paper and its Supplementary Information. Source data for the tumour-growth curves are provided with this paper. All data generated in this study are available from the corresponding authors on reasonable request. Source data are provided with this paper.

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

## Acknowledgements

The anti-NAP antibody was kindly provided by M. de Bernard (University of Padua, Padua, Italy). The protocol for transduction of mouse T cells was kindly provided by A. Mondino (San Raffaele Hospital, Milan, Italy). We also thank BioVis Platform (Uppsala University) for flow cytometry and imaging service, and A. Dimberg (Uppsala University) for critical reading of the manuscript. This work was supported by the Swedish Cancer Society (190184Pj, 190188Us); the Swedish Research Council (2019-01326); the Swedish Childhood Cancer Fund (PR2018-0127); the Sjöberg Foundation (2020-01-07-06); the Clas Groschinsky Foundation (M19359); the Erik, Karin and Gösta Selander Foundation (D.Y.); and the Göran Gustafsson Foundation (2003). J.M. was supported by a fellowship from the Chinese Scholarship Council (201406300037). M.R. was supported by the Swedish Childhood Cancer Fund (TJ2019-0014).

## Author contributions

C.J., J.M., M.R., D.Y. and M.E. conceived and designed the experiments. C.J., J.M. and M.R. performed the experiments and analysed the data. C.J., J.M., M.R., D.Y. and M.E. wrote the paper. All authors read and approved the final version of the manuscript.

## Funding

## Competing interests

D.Y. and M.E. are co-founders of Elicera Therapeutics, and have submitted a patent application (PCT/WO2018050225A1/2016) based on the findings of this work. The other authors declare no competing interests.

## Additional information

**Extended data** is available for this paper at https://doi.org/10.1038/s41551-022-00875-5.

**Correspondence and requests for materials** should be addressed to Di Yu or Magnus Essand.

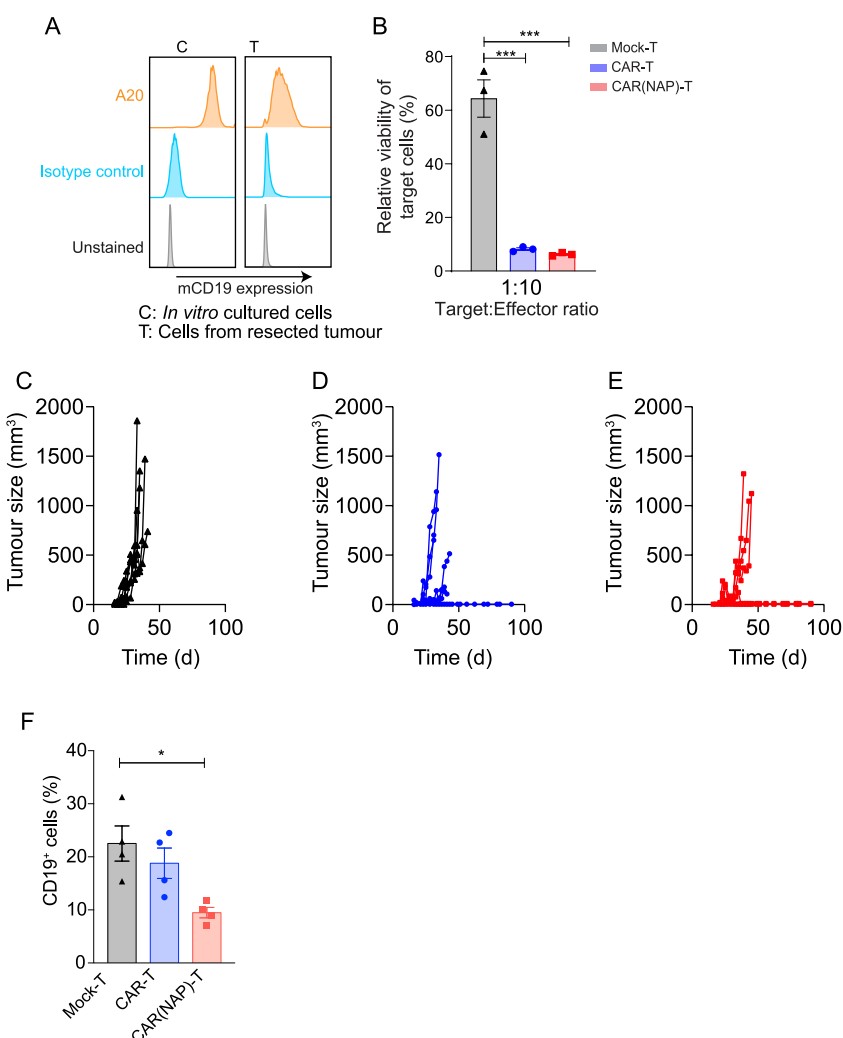

**Extended Data Fig. 1 | Therapeutic effect of mCD19-directed murine CAR T cells and CAR(NAP) T cells against mouse lymphoma (A20)** *in vitro* **and** *in vivo***.** (**a**) Expression level of mouse CD19 on A20 cells cultured *in vitro* (C) and isolated from resected subcutaneously growing A20 tumours (T). (**b**) The cytotoxic effect of engineered mCD19-targeted murine CAR T cells against *in vitro*-cultured A20. Error bars represent SEM (***: $P < 0.001$). (**c-e**) Tumour size of individual mice after treatment with (**C**) Mock T cells, (**D**) CAR T cell, and (**E**) CAR(NAP) T cell treatment. (**f**) B-cell (CD19+) count in peripheral blood in mice treated with engineered T cells (n = 4, *: P < 0.05). Precise *P*-values are reported in Supplementary Table 3.

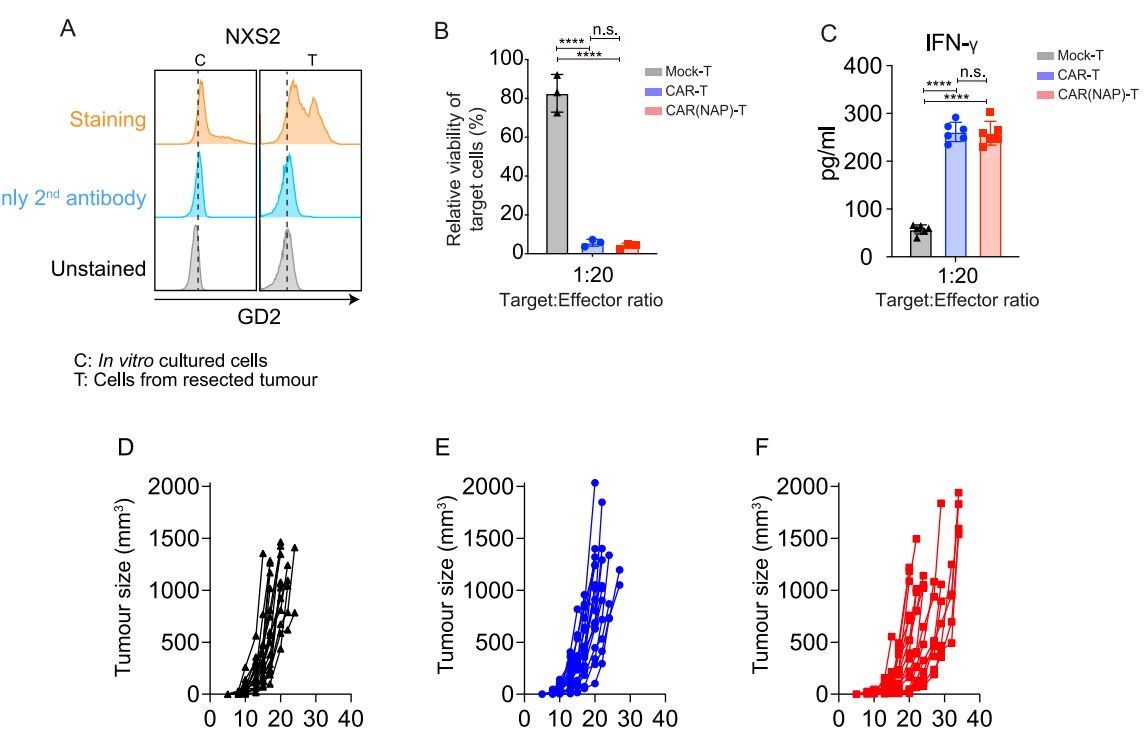

**Extended Data Fig. 2 | Therapeutic effect of GD2-directed murine CAR T cells and CAR(NAP) T cells against mouse neuroblastoma (NXS2)** *in vitro* **and** *in vivo*. (**a**) Expression level of GD2 on NXS2 cells cultured *in vitro* (C) and isolated form resected NXS2 subcutaneously growing tumours (T). (**b**) The cytotoxic effect of engineered GD2-targeted murine T cells against NXS2 cells *in vitro*, and (**c**) IFN-γ secretion by engineered T cells upon antigen encounter. Error bars represent SEM (n.s.: no statistical significance, ****: $P < 0.0001$). (**d-f**) Tumour size of individual mice after treatment with Mock T cells (**D**), CAR T cells (**E**), and CAR(NAP) T cells (**F**). Precise *P*-values are reported in Supplementary Table 3.

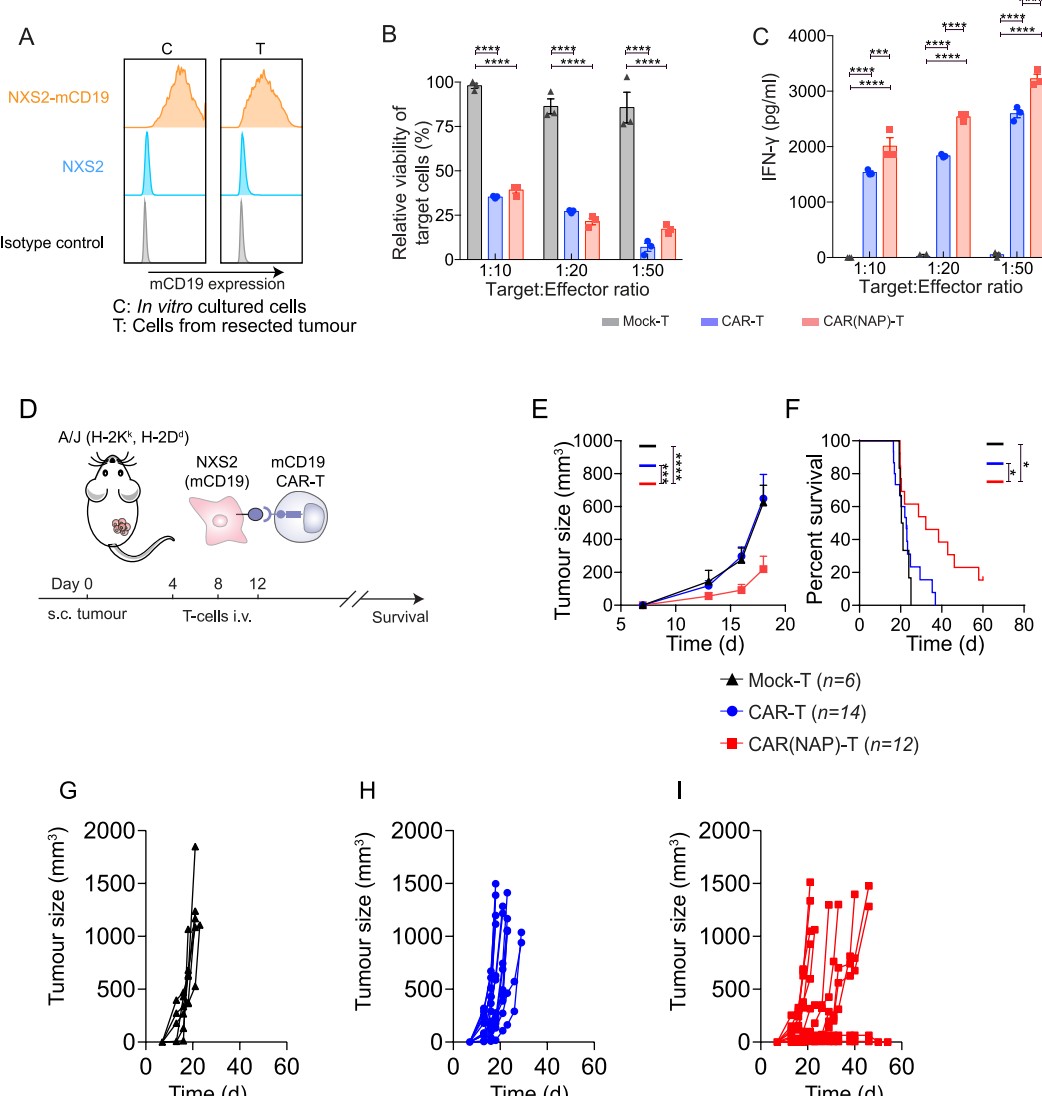

**Extended Data Fig. 3 | Therapeutic effect of mCD19-directed murine CAR T cells and CAR(NAP) T cells against NXS2-mCD19 *in vitro* and *in vivo*.** (**a**) Expression level of murine CD19 on wild-type NXS2 cells and NXS2 engineered to express murine CD19 (NXS2-mCD19), both cultured *in vitro* (C) and isolated from resected subcutaneous tumours (T). (**b**) The cytotoxicity of engineered mCD19-directed murine T cells against NXS2-mCD19 cells *in vitro*, and (**c**) IFN-γ secretion by engineered T cells upon antigen encounter. Error bars represent SEM. (**d**) Treatment schedule for the subcutaneous tumour model. (**e**) Tumour size (mean) and (**f**) mouse survival (Kaplan-Meier curve) after treatment. Groups were compared by using the log-rank test (*: P < 0.05, ***: P < 0.001, ****: P < 0.0001). (**g-i**) Tumour size of individual mice after treatment with (**G**) Mock T cells, (**H**) CAR T cells, and (**I**) CAR(NAP) T cells. Precise *P*-values are reported in Supplementary Table 3.

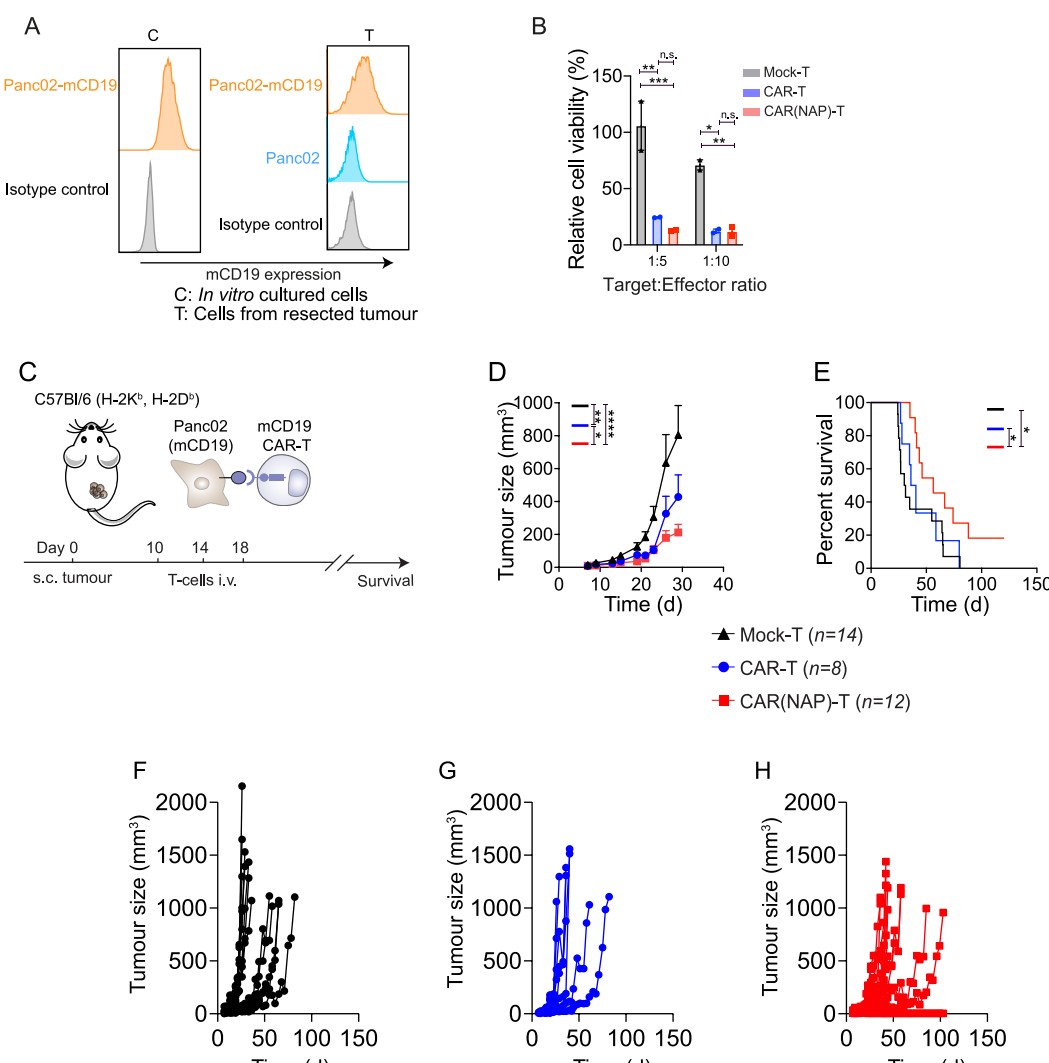

**Extended Data Fig. 4 | Therapeutic effect of mCD19-directed murine CAR T cells and CAR(NAP) T cells against Panc02-mCD19 *in vitro* and *in vivo*.** (**a**) Expression level of murine CD19 on wild-type Panc02 and Panc02 engineered to express murine CD19 (Panc02-mCD19), both cultured *in vitro* (C) and isolated from resected subcutaneous tumours (T). (**b**) The cytotoxicity of engineered murine T cells against Panc02-mCD19 *in vitro*. (**c**) Treatment schedule for the subcutaneous tumour model. (**d**) Tumour size (mean) and (**e**) mouse survival (Kaplan-Meier curve) after treatment. Groups were compared by using the log-rank test. Error bars represent SEM (n.s.: no statistical significance, *: P < 0.05, **: P < 0.01, ***: P < 0.001, ****: P < 0.0001). (**f-h**) Tumour size of individual mice after treatment with (**F**) Mock T cells, (**G**) CAR T cells, and (**H**) CAR(NAP) T cells. Precise *P*-values are reported in Supplementary Table 3.

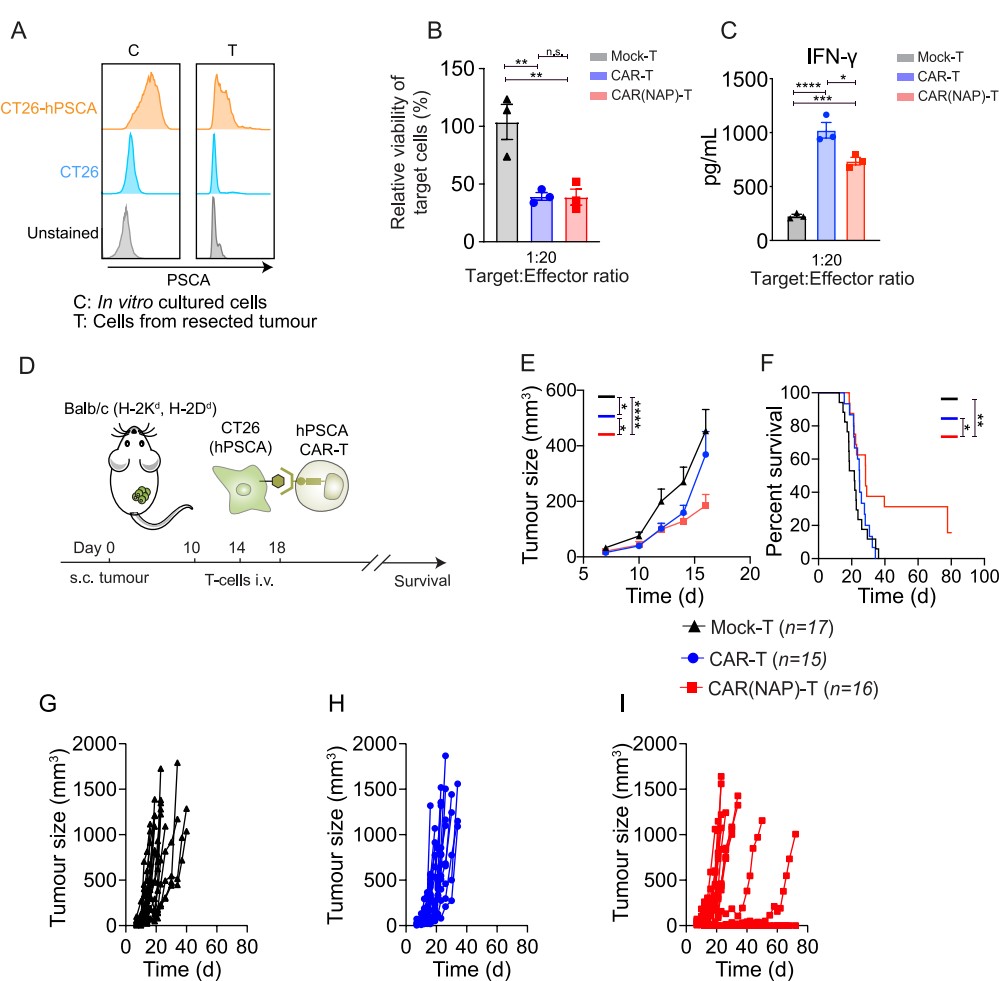

**Extended Data Fig. 5 | Therapeutic effect of hPSCA-directed murine CAR T cells and CAR(NAP) T cells against CT26-hPSCA *in vitro* and *in vivo*.** (**a**) Expression level of human prostate stem cell antigen (hPSCA) on wild-type CT26 and CT26 engineered to express human PSCA (CT26-hPSCA) cultured *in vitro*. (**b**) The cytotoxicity of engineered hPSCA-targeted murine T cells against CT26-hPSCA tumour cells *in vitro*, and (**c**) IFN-γ secretion by engineered T cells upon antigen encounter. Error bars represent SEM. (**d**) Treatment schedule for the subcutaneous tumours. (**e**) Tumour size (mean) and (**f**) mouse survival (Kaplan-Meier curve) after treatment. Groups were compared by using the log-rank test (n.s.: no statistical significance, *: P < 0.05, **: P < 0.01, ***: P < 0.001, ****: P < 0.0001). (**g-i**) Tumour size of individual mice after treatment with (**G**) Mock T cells, (**H**) CAR T cells, and (**I**) CAR(NAP) T cells. Precise *P*-values are reported in Supplementary Table 3.

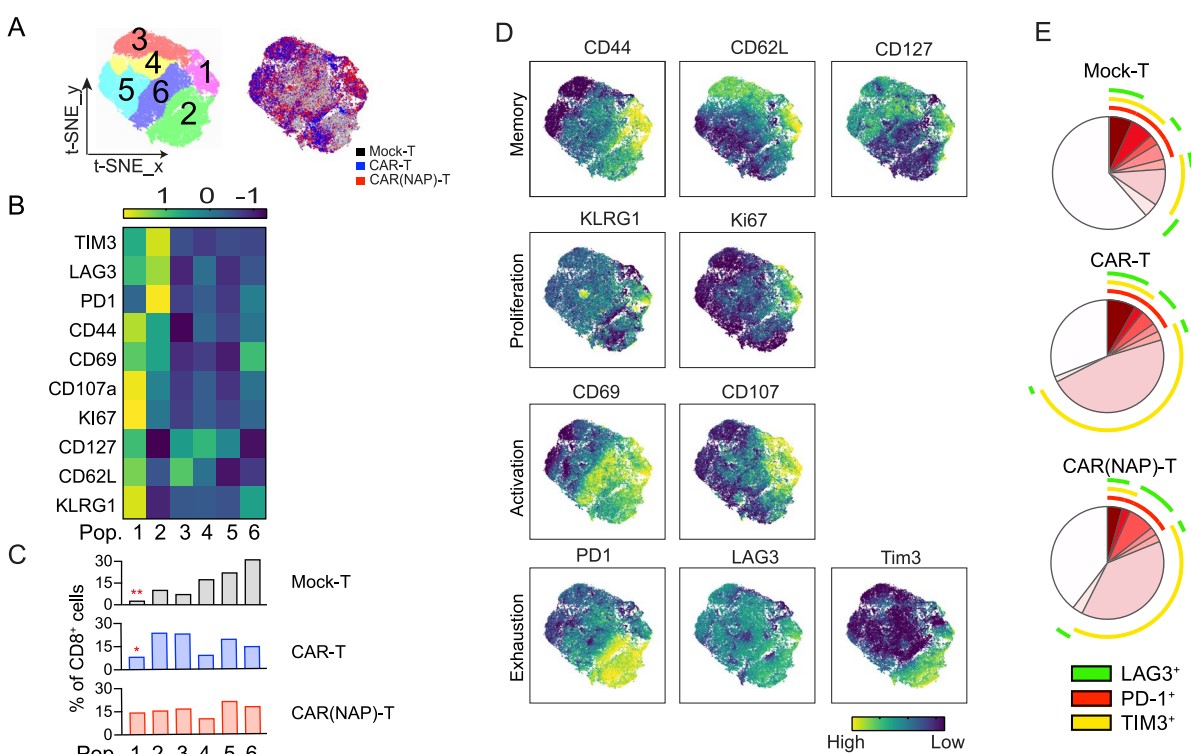

**Extended Data Fig. 6 | Characterization the tumour-infiltrating T cells in the NXS2-mCD19-OVA model in A/J mice after treatment.** (**a**) tSNE analysis and unsupervised clustering of pooled tumour-infiltrating CD8+ T cells revealed six populations (Pop1-6) clustered (left) based on expression of phenotypic markers and their distribution across treatment groups (right). (**b**) Median expression of each marker (Z-score-transformed) in Pop1-6. (**c**) Mean percentage of CD8+ T cells within Pop1-6. Asterisk indicating comparison to the CAR(NAP)-T group (*: $P < 0.05$, **: $P < 0.01$). (**d**) tSNE plots showing expression of surface markers on tumour-infiltrating CD8+ T cells. (**e**) Percentage of tumour-infiltrating CD8+ T cells with single, double, and triple positive expression of LAG3, PD-1, and TIM3. Precise $P$-values are reported in Supplementary Table 3.

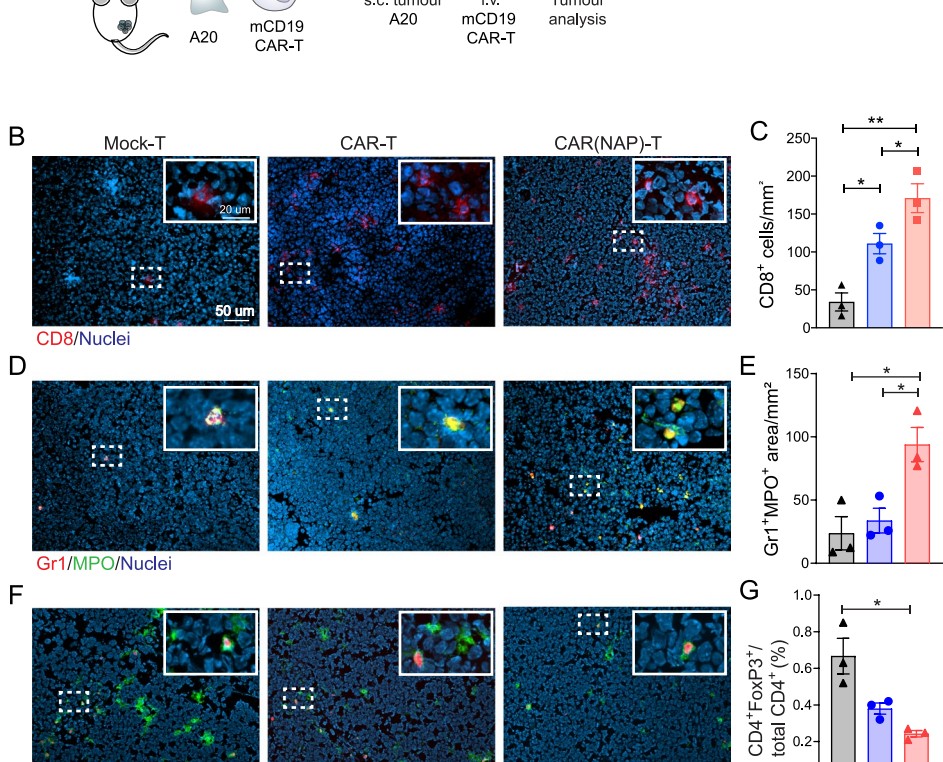

**Extended Data Fig. 7 | CAR(NAP) T cell treatment alters immune cell infiltration in the A20 lymphoma tumour model. (a)** Experimental set-up. (**b-g**) Immune cell infiltration in tumours harvested on day 25 after tumour cell implantation from different treatment groups, analysed by fluorescence staining. (**B**) Representative images and (**C**) quantification of tumour-infiltrating CD8$^+$ T cells. (**D**) Representative images and (**E**) quantification of Gr1$^+$MPO$^+$ neutrophils. (**F**) Representative images and (**G**) quantification of CD4$^+$FoxP3$^+$ T cells. Scale bars in all larger panels, 50 μm; scale bars in insets, 20 μm. Error bars represent SEM (*: $P < 0.05$, **: $P < 0.01$). Precise $P$-values are reported in Supplementary Table 3.

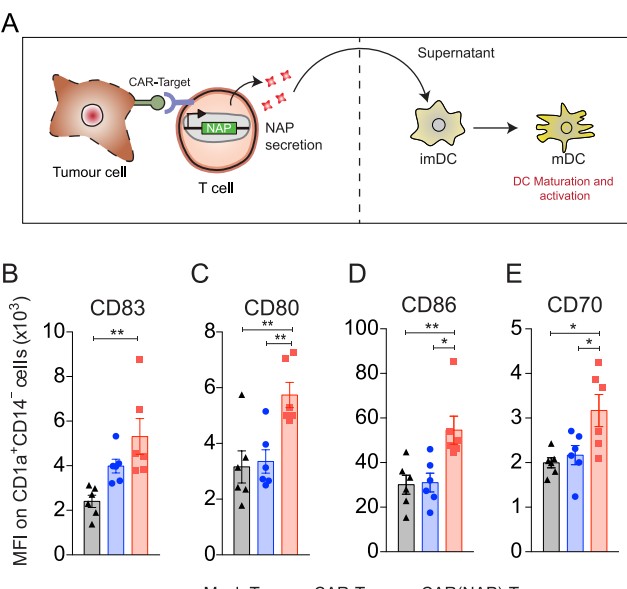

**Extended Data Fig. 8 | NAP secreted by activated CAR(NAP) T cells promotes monocyte-derived DC maturation and activation. (a**) Illustration of experimental design for assessment of DC maturation and activation *in vitro*. (**b-e**) Surface marker expression on DCs after 48 h culture in supernatants from co-cultures of engineered T cells and target tumour cells (5:1). All experiments were repeated and data were pooled. Error bars represent SEM (*: $P < 0.05$, **: $P < 0.01$). Precise *P*-values are reported in Supplementary Table 3.

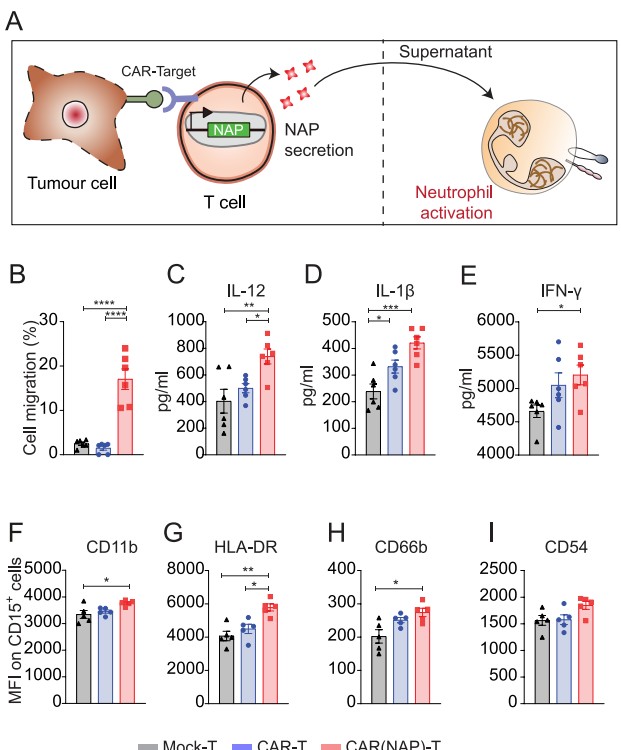

**Extended Data Fig. 9 | NAP secreted by activated CAR(NAP) T cells promotes neutrophil recruitment and activation.** (**a**) Overview of experimental design for assessing neutrophil recruitment and activation *in vitro*. (**b**) Percentage of neutrophil migration towards the supernatant from different co-cultures of engineered T cells and target tumour cells (5:1). (**c-e**) Cytokine levels (IL-12, IL-1β, and IFN-γ) in supernatant of activated neutrophils, determined by ELISA. (**f-i**) Cell surface markers on neutrophils (gated as CD15+ cells) after 48 h co-culture, assessed by flow cytometry. All experiments were repeated and data were pooled. Error bars represent SEM (*: $P < 0.05$, **: $P < 0.01$, ***: $P < 0.001$, ****: $P < 0.0001$). Precise *P*-values are reported in Supplementary Table 3.

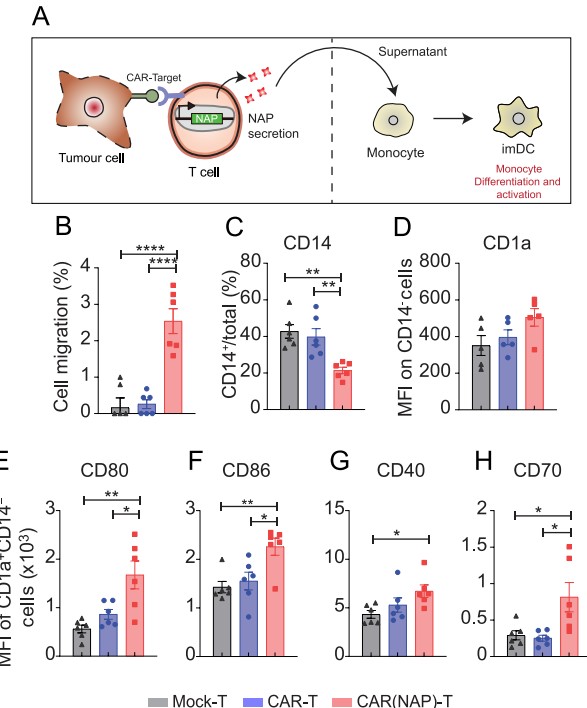

**Extended Data Fig. 10 | NAP secreted by activated CAR(NAP) T cells promotes monocyte differentiation and activation.** (**a**) Overview of experimental design for assessing monocyte differentiation and activation *in vitro*. (**b**) Percentage of monocyte migration towards the supernatant from different co-cultures of engineered T cells and target tumour cells. (**c**) Monocyte differentiation into immature DCs, presented as a percentage of CD14+ cells (monocytes) remaining after 4 d co-culture and (**d**) the expression of CD1a on CD14- (differentiated) cells. (**e-h**) Expression of DC activation markers on differentiated monocytes (gated as CD14-CD1a+). All experiments were repeated and data were pooled. Error bars represent SEM (**: *P* < 0.01). Precise *P*-values are reported in Supplementary Table 3.

# Reporting Summary

## Statistics

For all statistical analyses, confirm that the following items are present in the figure legend, table legend, main text, or Methods section.

| n/a | Confirmed | |
|---|---|---|
| ☐ | ☒ | The exact sample size (*n*) for each experimental group/condition, given as a discrete number and unit of measurement |
| ☐ | ☒ | A statement on whether measurements were taken from distinct samples or whether the same sample was measured repeatedly |
| ☐ | ☒ | The statistical test(s) used AND whether they are one- or two-sided<br>*Only common tests should be described solely by name; describe more complex techniques in the Methods section.* |
| ☐ | ☒ | A description of all covariates tested |
| ☐ | ☒ | A description of any assumptions or corrections, such as tests of normality and adjustment for multiple comparisons |
| ☐ | ☒ | A full description of the statistical parameters including central tendency (e.g. means) or other basic estimates (e.g. regression coefficient) AND variation (e.g. standard deviation) or associated estimates of uncertainty (e.g. confidence intervals) |
| ☐ | ☒ | For null hypothesis testing, the test statistic (e.g. *F*, *t*, *r*) with confidence intervals, effect sizes, degrees of freedom and *P* value noted<br>*Give P values as exact values whenever suitable.* |
| ☒ | ☐ | For Bayesian analysis, information on the choice of priors and Markov chain Monte Carlo settings |
| ☒ | ☐ | For hierarchical and complex designs, identification of the appropriate level for tests and full reporting of outcomes |
| ☒ | ☐ | Estimates of effect sizes (e.g. Cohen's *d*, Pearson's *r*), indicating how they were calculated |

*Our web collection on statistics for biologists contains articles on many of the points above.*

## Software and code

Policy information about availability of computer code

| Data collection | Flow cytometry: BD Diva; CytExpert;  BD FACSChorus; BD LSRFortessa<br>Microscopy: Zeiss AxioImager<br>Protein multiplexing imaging: BioRad Imager |
|---|---|
| Data analysis | Graphpad v6.07 - v9.3.1;<br>Nanostring: nSolver 4.0<br>GO analysis: Metascape<br>Flow: FlowJo v10.2 - v10.8.1 |

For manuscripts utilizing custom algorithms or software that are central to the research but not yet described in published literature, software must be made available to editors and reviewers. We strongly encourage code deposition in a community repository (e.g. GitHub). See the Nature Portfolio guidelines for submitting code & software for further information.

## Data

Policy information about availability of data

All manuscripts must include a data availability statement. This statement should provide the following information, where applicable:

- Accession codes, unique identifiers, or web links for publicly available datasets
- A description of any restrictions on data availability
- For clinical datasets or third party data, please ensure that the statement adheres to our policy

The main data supporting the results in this study are available within the paper and its Supplementary Information. Source data for the tumour-growth curves are provided with this paper. All data generated in this study are available from the corresponding authors on reasonable request.

# Field-specific reporting

Please select the one below that is the best fit for your research. If you are not sure, read the appropriate sections before making your selection.

☒ Life sciences ☐ Behavioural & social sciences ☐ Ecological, evolutionary & environmental sciences

For a reference copy of the document with all sections, see nature.com/documents/nr-reporting-summary-flat.pdf

# Life sciences study design

All studies must disclose on these points even when the disclosure is negative.

| | |
|---|---|
| Sample size | No sample-size calculations were performed. Sample size was based on previous experience and pilot studies, and each experiment was repeated at least once. |
| Data exclusions | Mice injected with tumour cells and that did not show tumour growth were not included. Mice that died for unknown reasons were excluded. No data were excluded from the analyses. |
| Replication | The animal studies were performed at least twice, and the data were pooled. In vitro data were generated from at least 3 donors, as detailed in the relevant figure captions. |
| Randomization | All groups of mice were age-matched. The mice were randomized prior to treatment, without knowledge of tumor burden. |
| Blinding | Blinding was irrelevant to the study. All mice experiments were carried out by researchers who also prepared the CAR-T cells before administration into mice. |

# Reporting for specific materials, systems and methods

We require information from authors about some types of materials, experimental systems and methods used in many studies. Here, indicate whether each material, system or method listed is relevant to your study. If you are not sure if a list item applies to your research, read the appropriate section before selecting a response.

### Materials & experimental systems

| n/a | Involved in the study |
|---|---|
| ☐ | ☒ Antibodies |
| ☐ | ☒ Eukaryotic cell lines |
| ☒ | ☐ Palaeontology and archaeology |
| ☐ | ☒ Animals and other organisms |
| ☐ | ☒ Human research participants |
| ☒ | ☐ Clinical data |
| ☒ | ☐ Dual use research of concern |

### Methods

| n/a | Involved in the study |
|---|---|
| ☒ | ☐ ChIP-seq |
| ☐ | ☒ Flow cytometry |
| ☒ | ☐ MRI-based neuroimaging |

## Antibodies

| | |
|---|---|
| Antibodies used | All antibodies are listed in Supplementary Table 1. |
| Validation | Antibody validation was performed by the relevant supplier. |

## Eukaryotic cell lines

Policy information about cell lines

| | |
|---|---|
| Cell line source(s) | Daudi, BC-3, A20 and CT26 cells were procured from ATCC; NXS2 was a gift from Holger N. Lode, University of Greifswald; Panc02 was a gift from Rainer Heuchel, Karolinska Institute. Human PBMCs (source of T cells) were freshly isolated from buffycoat. Mouse splenocytes (source of T cells) were freshly isolated from mouse spleens. The Gryphon retroviral packaging cell line was procured from Allele Biotechnology, San Diego, CA. The 293T lentiviral/retroviral vector packaging cell line was procured from ATCC. |
| Authentication | The cell lines purchased from ATCC were not authenticated. Murine cell lines from C57bl/6 were confirmed with the strain, but cannot be authenticated by STR. |
| Mycoplasma contamination | All cell lines tested negative for mycoplasma, using a Lonza kit. |

| Commonly misidentified lines (See ICLAC register) | No commonly misidentified cell lines were used (according to ICLAC version 11; checked on 07 February 2022). |
|---|---|

## Animals and other organisms

Policy information about studies involving animals; ARRIVE guidelines recommended for reporting animal research

| Laboratory animals | 6–8-week-old female C57Bl/6NRj and Balb/c from Taconic Denmark, and 6–8-week old female A/J mice from Envigo, the Netherlands. |
|---|---|
| Wild animals | The study did not involve wild animals. |
| Field-collected samples | The study did not involve samples collected from the field. |
| Ethics oversight | The Uppsala Research Animal Ethics Committee approved all animal studies (N164/15; N185/16; 5.8.18-19434/2019). |

Note that full information on the approval of the study protocol must also be provided in the manuscript.

## Human research participants

Policy information about studies involving human research participants

| Population characteristics | The human buffy coats obtained from healthy blood donors had been anonymized. |
|---|---|
| Recruitment | Peripheral blood mononuclear cells were isolated by Ficoll-Paque (GE Healthcare Life Science, Uppsala, Sweden) from fresh buffy coats of healthy anonymized donors, collected at the Blood Centre at Uppsala University Hospital. |
| Ethics oversight | Because the samples had been anonymized, an ethical permit was not required. |

Note that full information on the approval of the study protocol must also be provided in the manuscript.

## Flow Cytometry

### Plots

Confirm that:

☐ The axis labels state the marker and fluorochrome used (e.g. CD4-FITC).

☐ The axis scales are clearly visible. Include numbers along axes only for bottom left plot of group (a 'group' is an analysis of identical markers).

☐ All plots are contour plots with outliers or pseudocolor plots.

☒ A numerical value for number of cells or percentage (with statistics) is provided.

### Methodology

| Sample preparation | For the cell-culture experiments: suspension cells were harvested directly, washed with PBS and resuspended in ca. 200 µL of PBS containing the indicated antibody mixture (prepared as a master-mix solution).

For samples with intracellular-staining steps: cells were first stained with surface markers, then permeabilized with BDperm buffer (BD Biosciences) and additionally stained with antibodies targeting an intracellular marker (such as IL-2).

For samples with intra-nucleus staining steps: cells were first stained with surface markers, then permeabilized with True-Nuclear Transcription Factor Buffer set (Biolegend), then washed, and stained with additional antibodies (such as T-bet).

For NAP-expression detection: cells were treated with Brefeldin A (BD GolgiPlug, BD BioSciences) before staining.

For the in vivo experiments: The tumour samples were collected and enzymatically digested (Liberase, Roche) into single-cell suspensions. CD45+ cells were bead-isolated (Miltenyi Biotec) and stained with the appropriate antibody mixture. |
|---|---|
| Instrument | BD Canto II; BD Melody; CytoFLEX S; CytoFLEX XL. |
| Software | BD Diva; CytExpert; BD FACSChorus. |
| Cell population abundance | At least 10,000 cells were recorded for CD3+ for T cells; at least 1x10^6 for alive CD45+ cells were recorded when analysing tumour-infiltrating cells; otherwise, at least 10,000 cells in the FSC/SSC gate were recorded. |
| Gating strategy | Preliminary FSC-A/SSC-A gates were used on morphology and FSC-A/FSC-H for singlets, then followed by Zombie Aqua live/dead gating. Further gating for each population are detailed in Methods and in the relevant figure captions. |

☒ Tick this box to confirm that a figure exemplifying the gating strategy is provided in the Supplementary Information.

