## [Peer Review File · Nature Biomedical Engineering]

CAR T cells expressing a bacterial virulence factor trigger potent bystander antitumour responses in solid cancers

Corresponding author: Di Yu

Editorial note

This document includes relevant written communications between the manuscript's corresponding author and the editor and reviewers of the manuscript during peer review. It includes decision letters relaying any editorial points and peer-review reports, and the authors' replies to these (under 'Rebuttal' headings). The editorial decisions are signed by the manuscript's handling editor, yet the editorial team and ultimately the journal's Chief Editor share responsibility for all decisions.

Any relevant documents attached to the decision letters are referred to as **Appendix #**, and can be found appended to this document. Any information deemed confidential has been redacted or removed. Earlier versions of the manuscript are not published, yet the originally submitted version may be available as a preprint. Because of editorial edits and changes during peer review, the published title of the paper and the title mentioned in below correspondence may differ.

Correspondence

Mon 14 Jun 2021

Decision on Article nBME-21-1069

Dear Dr Yu,

Thank you again for submitting to *Nature Biomedical Engineering* your manuscript, "Expression of a pathogenic virulence factor enhances the efficacy of CAR-T cell therapy against solid tumors". The manuscript has been seen by three experts, whose reports you will find at the end of this message. You will see that the reviewers have good words for the work, and that they raise a number of technical criticisms that we hope you will be able to address. In particular, we would expect that a revised version of the manuscript provides:

* Evidence and discussion of the in vivo immunogenicity, potential toxicity, and extent of persistence of the CAR-T cells expressing the bacterial protein, as per the comments of all reviewers.

* Thorough performance comparisons across groups (with detailed reporting of the statistical-significance tests carried out). Also, for all tumour-growth curves, please provide the individual-mouse data (in the Supplementary Information, or as overlaid curves in the main figures).

When you are ready to resubmit your manuscript, please upload the revised files, a point-by-point rebuttal to the comments from all reviewers, the (revised, if needed) reporting summary, and a cover letter that explains the main improvements included in the revision and responds to any points highlighted in this decision.

Please follow the following recommendations:

* Clearly highlight any amendments to the text and figures to help the reviewers and editors find and understand the changes (yet keep in mind that excessive marking can hinder readability).- * If you and your co-authors disagree with a criticism, provide the arguments to the reviewer (optionally, indicate the relevant points in the cover letter).
- * If a criticism or suggestion is not addressed, please indicate so in the rebuttal to the reviewer comments and explain the reason(s).
- * Consider including responses to any criticisms raised by more than one reviewer at the beginning of the rebuttal, in a section addressed to all reviewers.
- * The rebuttal should include the reviewer comments in point-by-point format (please note that we provide all reviewers will the reports as they appear at the end of this message).
- * Provide the rebuttal to the reviewer comments and the cover letter as separate files.

We hope that you will be able to resubmit the manuscript within 20 weeks from the receipt of this message. If this is the case, you will be protected against potential scooping. Otherwise, we will be happy to consider a revised manuscript as long as the significance of the work is not compromised by work published elsewhere or accepted for publication at *Nature Biomedical Engineering*.

We hope that you will find the referee reports helpful when revising the work, which we look forward to receive. Please do not hesitate to contact me should you have any questions.

Best wishes,

Pep

Pep Pàmies
Chief Editor, Nature Biomedical Engineering

Reviewer #1 (Report for the authors (Required)):

The work by Jin and colleagues is a rigorous test of whether expression of NAP derive from *H. pylori* would increase potency of CAR T. The work is technically well done. Major strengths of the manuscript are the extensive characterization in both hematologic and solid tumor models. In vivo modeling is done in syngeneic mice. A major issue for the field is that CAR T cells have to date not shown to be effective for eliciting endogenous immunity against other antigens in the TME beyond the CAR T-cell target. Here the author's show that CAR T cells encoding NAP in mouse T cells and human T cells have effects through dendritic cells and other cells the microenvironment. The authors conclude with a demonstration that the same approach works in human T cells but they do not model this in immunodeficient mice. This is reasonable because the NSG mouse models would not be expected to mount an immune response through recruitment of endogenous immunity.

The major issue is that in the Discussion the author should add more to the primary limitation of this approach, which is whether immunity either humoral or cellular will prevent or limit the duration of efficacy with the expression of this bacterial protein? Also, authors should comment on whether there is pre-existing immunity in humans to NAP.

Authors should reference some literature where acquired immunity to engineered T cells has been encountered as a limitation:

1. Riddell SR, Elliott M, Lewinsohn DA, Gilbert MJ, Wilson L, Manley SA, Lupton SD, Overell RW, Reynolds TC, Corey L, and Greenberg PD. T-cell mediated rejection of gene-modified HIV-specific cytotoxic T lymphocytes in HIV-infected patients. *NatMed*. 1996;2(2):216-23.
2. Stripecke R, del Carmen Villacres M, Skelton D, Satake N, Halene S, and Kohn D. Immune response to

green fluorescent protein: implications for gene therapy. *Gene therapy*. 1999;6(7):1305-12.

3. Maus MV, Haas AR, Beatty GL, Albelda SM, Levine BL, Liu X, Zhao Y, Kalos M, and June CH. T Cells Expressing Chimeric Antigen Receptors Can Cause Anaphylaxis in Humans. *Cancer Immunology Research*. 2013;1:26-31.

Reviewer #2 (Report for the authors (Required)):

Chuan Jin and colleagues demonstrate that arming CAR-Ts to express *H. pylori* NAP induces bystander immunity via epitope spreading and the system is independent of tumor types and target antigens. The study is interesting and the conclusions are supported by strong data, obtained both in vitro and in vivo. However, I have some comments:

1. In Figure 1D the statistical significance of the difference in tumor size between samples exposed to CAR-T and those exposed to CAR(NAP)-T is not reported, even if in the text authors state that: "only CAR(NAP)-Ts significantly controlled tumor growth and prolonged survival of tumor-bearing mice in both models".
2. Referring to all experiments where tumor size was measured, was the statistical significance calculated at any time point or only at the end?
3. Figure 2H: what is the advantage of using NXS2-mCD19-OVA, in which OVA was introduced as a non-targeted bystander antigen, rather than using GD2 as non-targeted bystander antigen?
4. Why did the authors evaluate the infiltration of the immune cells in A20 tumor model by immunofluorescence (Figure S8) and not by FACS as they did in Figure 3? And why they did not evaluate the same immune cell populations?
5. Referring to Figure 3 authors state that the cell type profile obtained by gene expression analysis (panels F-H) was validated also by FACS (I-K). However, increase in cytotoxic NK cells is evidenced by gene expression and not by FACS and vice versa for antigen presenting DCs.
6. Since the analysis shown in Figure 3L did not reveal a robust level of Th1 cytokines, authors could reinforce the conclusion by quantifying the intra-cyto IFN γ in lymphocytes isolated from the tumor or local lymph nodes.
7. Figure S1D-E-F: to verify the induction of luciferase, authors implanted human cancer cells, expressing or not CD19. It is not clear to me if human CD19-targeting CAR-T cells or murine CD19-targeting CAR-T cells have been used. Moreover, why did the authors not implanted NXS2, expressing or not mCD19, rather than human Daudi (CD19+) and BC-3 (CD19-)?
8. How do the authors explain the different expression of the antigen between cultured in vitro cells and cells isolated from the tumor (Figure S2, S3, S4, S5 A)?
9. Why did the author choose different target:effector ratios in the experiments shown in Figure S2, S3, S4, S5 and S6 B?
10. In Figure S6A is not reported the expression of the antigen by cells isolated from resected tumors.
11. The reduced released of IFN γ in CAR(NAP)-T with respect to CAR-T (Figure S6C), is unexpected. Authors should provide a possible explanation.
12. Figure S7, it is not clear for what purpose were used either splenocytes or tumor-infiltrating CD8+ T cells? Legend must include more details on this matter.
13. Since CD19 is expressed by normal B cells, what about the impact on these cells in vitro and in vivo by the exposure/administration of CD19-targeted CAR-T and CAR(NAP)-T? Moreover at least one experiment by using as target cells unrelated and normal cells could be shown.

Reviewer #3 (Report for the authors (Required)):

The authors present a novel approach to enhancing CAR T cell efficacy through the additional introduction of the gene encoding H pylori neutrophil activating protein (NAP) which in turn allows the CAR T cell to introduce NAP to the tumor microenvironment leading to the recruitment of endogenous antitumor effectors including T cells as well as induction of DC maturation which in turn could enhance cross presentation of other tumor antigens to endogenous T cells. The authors demonstrate that these NAP secreting CAR T cells enhance tumor eradication in syngeneic tumor models and further demonstrate that these differences are due in part to modulation of the TME including DC maturation and recruitment of endogenous tumor specific T cells. The authors conclude that this is a proof of principle that bacterial derived proinflammatory factors may be a novel approach to enhance the efficacy of tumor targeted T cells in the context of adoptive cell therapies of cancer. The manuscript is well written and the data largely support the authors' conclusions. However, there are concerns regarding the approach and whether similar results have been demonstrated using other proinflammatory molecules.

Critiques

1. A primary concern regarding this approach is the fact that similar outcomes have been demonstrated in the context of, for example, CAR T cells modified to secrete cytokines (i.e. IL-18), or express ligands which enhance DC maturation (i.e. CD40L). Both examples use non-immunogenic reagents. To this end, the work would seem to generate similar outcomes just using a different, but bacterial, proinflammatory molecule (NAP).
2. The authors note in the discussion that this approach may be limited due to the bacterial origin of NAP which quite reasonably would be immunogenic and in fact many patients may potentially already harbor neutralizing antibodies which could impair the efficacy of this approach. Have the authors immunized mice with NAP and then see if these CAR T cells still function well?
3. The authors have not fully studied the safety of this approach especially in the setting of syngeneic tumor models and in the context of systemic tumors. Post mortem analyses of treated mice to assess for systemic inflammation would be helpful. Further does NAP impact the murine immune effectors to a similar degree as in the context of a human immune system?
4. Have the authors looked into the long term persistence of the CAR(NAP) T cells? This is highly relevant given the clinical relevance of CAR T cell persistence and even more so if NAP induces T cell mediated immune responses which could lead to rapid elimination of the CAR T cells. This could readily be explored in the syngeneic mouse models wherein CAR T cell persistence could be evaluated as well as assessment of whether endogenous T cells demonstrate cytotoxicity to CAR(NAP) T cells versus CAR T cells.
5. Some of the presented data does not support the authors' conclusions including figure 1E wherein there is no statistical difference between the CAR-T and CAR(NAP) T cell treated mice in survival, and all long term surviving mice were resistant to tumor rechallenge. It is also curious that NSX2 tumor models, the overall survival advantage is small and there are no long term surviving mice.

Fri 04 Feb 2022

Decision on Article nBME-21-1069A

Dear Dr Yu,

Thank you for your patience in waiting for the feedback on your revised manuscript, "Expression of a pathogenic virulence factor enhances the efficacy of CAR-T cell therapy against solid tumors". Having consulted with Reviewers #2 and #3 (whose comments you will find at the end of this message), I am pleased to write that we shall be happy to publish the manuscript in *Nature Biomedical Engineering*, provided that the points specified in the attached instructions file are addressed.

When you are ready to submit the final version of your manuscript, please upload the files specified in the instructions file.

For primary research originally submitted after December 1, 2019, we encourage authors to take up transparent peer review. If you are eligible and opt in to transparent peer review, we will publish, as a single supplementary file, all the reviewer comments for all the versions of the manuscript, your rebuttal letters, and the editorial decision letters. **If you opt in to transparent peer review, in the attached file please tick the box 'I wish to participate in transparent peer review'; if you prefer not to, please tick 'I do NOT wish to participate in transparent peer review'**. In the interest of confidentiality, we allow redactions to the rebuttal letters and to the reviewer comments. If you are concerned about the release of confidential data, please indicate what specific information you would like to have removed; we cannot incorporate redactions for any other reasons.

More information on transparent peer review is available.

Best wishes,

Pep

Pep Pàmies
Chief Editor, Nature Biomedical Engineering

P.S. Nature Research journals encourage authors to share their step-by-step experimental protocols on a protocol-sharing platform of their choice. Nature Research's Protocol Exchange is a free-to-use and open resource for protocols; protocols deposited in Protocol Exchange are citable and can be linked from the published article. More details can be found at www.nature.com/protocolexchange/about.

Reviewer #2 (Report for the authors (Required)):

Besides accurately addressing all my issues, authors performed new experiments to verify whether pre-existing antibodies against NAP affect the therapeutic efficacy of CAR-T cells armed to express NAP. The evidence that the anti-NAP immunity does not interfere with the CAR-T therapy is extremely important considering that the *H. pylori* infection is still widespread, and NAP is a major antigen.

Moreover, a new set of experiments was performed to rule out any possible toxic effect resulting from the treatment and to evaluate the long-term CAR-T persistence. Authors revealed that both CAR-T and CAR(NAP)-T were detectable in about 50% of the mice up to 70 days after administration and repeated treatment with CAR(NAP)-T did not result in elevated toxicity compared to conventional CAR-T when assessing systemic cytokine release or body weight.

Collectively, these new data significantly improved the quality of the manuscript but, most of all greatly enhanced the translational potential of this strategy for targeting solid tumors.

Reviewer #3 (Report for the authors (Required)):

Revised manuscript suitable for publication.

Rebuttal 1

Dear Editor and Reviewers

We appreciate your positive feedback on our manuscript (nBME-21-1069) entitled “Expression of a pathogenic virulence factor enhances the efficacy of CAR-T cell therapy against solid tumors”. We have performed new animal experiments that specifically address the questions raised by all reviewers and editor. Please find our revised manuscript with changes/additional texts **highlighted in red**, as well as point-to-point answers (in blue) to each of your comments below.

We believe that the comments from editor and reviewers were highly relevant and have helped us to significantly improve the manuscript. We sincerely hope that it can now be regarded acceptable for publication in Nature Biomedical Engineering.

Responses to all reviewers and editor:

* Evidence and discussion of the in vivo immunogenicity, potential toxicity, and extent of persistence of the CAR-T cells expressing the bacterial protein, as per the comments of all reviewers.

We have performed new animal experiment to evaluate the therapeutic effects of conventional CAR-T and CAR(NAP)-T in tumor-bearing mice with pre-existing anti-NAP antibodies, wherein mice were vaccinated with an adenoviral vector expressing NAP before tumor cell implantation. In repeated experiments we found the overall survival was not affected by the pre-existing anti-NAP immunity. The new data are presented in the new submitted manuscript as a new Supplementary Fig. S8.

We also performed other animal experiments where toxicity and long-term CAR-T persistence were followed. We can see that both CAR-T and CAR(NAP)-T were detectable in about 50% of the mice up to 70 days after administration (Supp Fig. S7). We did not detect a spike in cytokines in the serum related to Cytokine release syndrome after CAR(NAP)-T treatment (Supp Fig. S9). In addition, we also report the mouse body weight in each treatment experiment as safety evaluation (Supp Fig. S10).

* Thorough performance comparisons across groups (with detailed reporting of the statistical-significance tests carried out). Also, for all tumour-growth curves, please provide the individual-mouse data (in the Supplementary Information, or as overlaid curves in the main figures).

We have now revised the manuscript and reported all statistical method used and the significance for each figure. Individual mouse tumor growth curves are now provided in the corresponding supplementary information.

Response to specific questions to each reviewer

Reviewer #1 (Report for the authors (Required)):

The work by Jin and colleagues is a rigorous test of whether expression of NAP derive from *H. pylori* would increase potency of CAR T. The work is technically well done. Major strengths of the manuscript are the extensive characterization in both hematologic and solid tumor models. In vivo modeling is done in syngeneic mice. A major issue for the field is that CAR T cells have to date not

shown to be effective for eliciting endogenous immunity against other antigens in the TME beyond the CAR T-cell target. Here the author's show that CAR T cells encoding NAP in mouse T cells and human T cells have effects through dendritic cells and other cells the microenvironment. The authors conclude with a demonstration that the same approach works in human T cells but they do not model this in immunodeficient mice. This is reasonable because the NSG mouse models would not be expected to mount an immune response through recruitment of endogenous immunity.

The major issue is that in the Discussion the author should add more to the primary limitation of this approach, which is whether immunity either humoral or cellular will prevent or limit the duration of efficacy with the expression of this bacterial protein? Also, authors should comment on whether there is pre-existing immunity in humans to NAP.

Authors should reference some literature where acquired immunity to engineered T cells has been encountered as a limitation:

1. Riddell SR, Elliott M, Lewinsohn DA, Gilbert MJ, Wilson L, Manley SA, Lupton SD, Overell RW, Reynolds TC, Corey L, and Greenberg PD. T-cell mediated rejection of gene-modified HIV-specific cytotoxic T lymphocytes in HIV-infected patients. *NatMed*. 1996;2(2):216-23.
2. Stripecke R, del Carmen Villacres M, Skelton D, Satake N, Halene S, and Kohn D. Immune response to green fluorescent protein: implications for gene therapy. *Gene therapy*. 1999;6(7):1305-12.
3. Maus MV, Haas AR, Beatty GL, Albelda SM, Levine BL, Liu X, Zhao Y, Kalos M, and June CH. T Cells Expressing Chimeric Antigen Receptors Can Cause Anaphylaxis in Humans. *Cancer Immunology Research*. 2013;1:26-31.

We greatly appreciate the reviewers positive feedback and have now performed new animal experiments addressing the questions regarding pre-existing anti-NAP immunity. As all reviewers mentioned this, we have summarized our response in the section above "Response to all reviewers". We hope that the new data are sufficient.

We have also added the references mentioned.

Reviewer #2 (Report for the authors (Required)):

Chuan Jin and colleagues demonstrate that arming CAR-Ts to express H. pylori NAP induces bystander immunity via epitope spreading and the system is independent of tumor types and target antigens.

The study is interesting and the conclusions are supported by strong data, obtained both in vitro and in vivo. However, I have some comments:

1. In Figure 1D the statistical significance of the difference in tumor size between samples exposed to CAR-T and those exposed to CAR(NAP)-T is not reported, even if in the text authors state that: "only CAR(NAP)-Ts significantly controlled tumor growth and prolonged survival of tumor-bearing mice in both models".

We thank the reviewer's carefully pointing out this mistake. There is no statistical difference between the CAR-T and CAR(NAP)-T groups; or between the CAR-T and Mock-T groups. We have added the statistics in the new submitted figure and rephrased the text in the results.

2. Referring to all experiments where tumor size was measured, was the statistical significance calculated at any time point or only at the end?

The statistics were calculated when the first mice had to be sacrificed in the experiment (thus, the last day reported in the graphs showing mean tumors sizes). We have now clarified this in the Figure Legends.

3. Figure 2H: what is the advantage of using NXS2-mCD19-OVA, in which OVA was introduced as a non-targeted bystander antigen, rather than using GD2 as non-targeted bystander antigen?

We thank the reviewer for this comment. OVA is a protein antigen while GD2 is a disialoganglioside. Thus, using OVA as a bystander antigen allowed us to use established molecular tools, such as tetramer staining to investigate OVA-specific T cell response. In addition, protein-based antigen also allows us to detect a diversified T cell response against different TCR epitopes using peptide pool and ELISA as read-out (Fig 2G). Lack of specific detection method obstructed us from using GD2 as a bystander antigen.

4. Why did the authors evaluate the infiltration of the immune cells in A20 tumor model by immunofluorescence (Figure S8) and not by FACS as they did in Figure 3? And why they did not evaluate the same immune cell populations?

The reason for using immunofluorescence (now Fig S14) or FACS in Fig 3 is mainly based on the material available. A20 tumors can easily be cryo-sectioned while for NXS2 tumors it is more difficult to obtain good cryosection and therefore add an extra difficulty to sample analysis. In addition, we used different methods to detect immune cell infiltration as we believe that it increases the confidence of the reported data if different methods yield similar results.

5. Referring to Figure 3 authors state that the cell type profile obtained by gene expression analysis (panels F-H) was validated also by FACS (I-K). However, increase in cytotoxic NK cells is evidenced by gene expression and not by FACS and vice versa for antigen presenting DCs.

We thank the reviewer for pointing out this mistake. We have now re-phrased the sentence in our new submitted manuscript.

6. Since the analysis shown in Figure 3L did not reveal a robust level of Th1 cytokines, authors could reinforce the conclusion by quantifying the intra-cyto IFN γ in lymphocytes isolated from the tumor or local lymph nodes.

We greatly appreciate this valid point by the reviewer and have performed intracellular staining of IFN-gamma and IL-2 on tumor-infiltrating CD3+ T cells. Higher percentage of IFN-gamma and IL-2 producing T cells were found in tumor after CAR(NAP)-T treatment, and the new data are also shown in the new submitted manuscript as Fig. 3M, N.

7. Figure S1D-E-F: to verify the induction of luciferase, authors implanted human cancer cells, expressing or not CD19. It is not clear to me if human CD19-targeting CAR-T cells or murine CD19-targeting CAR-T cells have been used. Moreover, why did the authors not implanted NXS2, expressing or not mCD19, rather than human Daudi (CD19+) and BC-3 (CD19-)?

We thank the reviewer for this comment and have now clarified in the figure legend of Fig S1 that it was human CAR-T injected into nude mice with human cancers (Daudi or BC-3). We did not actively disregard to use NXS2 and NXS2-mCD19 cells, it is rather a result of how the project developed. This experiment was performed many years ago when we started the project. At that time, our focus was on proof-of-concept experiments to see whether our idea worked or not. Daudi and BC-3 cells, as well as the nLuc-expressing CAR-T were injected intravenously, which we believe is a “cleaner” experimental setup compared to using the NXS2 model that involves subcutaneous tumor development. We believe that our model can answer the question whether the transgene can be inducible expressed or not, and facilitate our further development of CAR(NAP)-T constructs.

8. How do the authors explain the different expression of the antigen between cultured *in vitro* cells and cells isolated from the tumor (Figure S2, S3, S4, S5 A)?

We thank the reviewer for this question. Cells engineered to ectopically express a protein can gradually reduce expression and even after some time lose the protein expression if a selection pressure is not presented which is the case *in vivo*, unlike cells cultured *in vitro* in the presence of puromycin (a selection marker co-expressed with the antigen during engineering). This phenomenon has also been reported by others (PMID: 16140581), and is most likely due to epigenetic silencing of the promoter. However, our key message is to convey that we do see expression of the CAR-target antigens in the different models examined, thus ruling out that lack of efficacy of the conventional CAR-T is due to loss of the antigen.

9. Why did the author choose different target:effector ratios in the experiments shown in Figure S2, S3, S4, S5 and S6 B?

The reason we use different E:T ratio is due to that different CARs have different efficacy and that the different retrovirus batches resulted in different transduction efficacy. Otherwise, there was no specific reason to select different target:effector ratios.

10. In Figure S6A is not reported the expression of the antigen by cells isolated from resected tumors.

We thank the reviewer for observing this. Due to miscommunication, the resected tumors were not saved. We have now performed new experiment assessing the human PSCA expression on resected tumor, and the new data are embedded in the updated Fig. S6A.

11. The reduced released of IFN γ in CAR(NAP)-T with respect to CAR-T (Figure S6C), is unexpected. Authors should provide a possible explanation.

We agree with the reviewer that it is unexpected that the hPSCA-directed CAR(NAP)-T yielded somewhat lower IFN-g release than the conventional hPSCA-directed CAR-T. We have no plausible explanation as to why. We would like to point that hPSCA-directed CAR(NAP)-T release significantly higher IFN-g release than Mock-T, illustrating the function of CAR(NAP)-T *in vitro*.

12. Figure S7, it is not clear for what purpose were used either splenocytes or tumor-infiltrating CD8+ T cells? Legend must include more details on this matter.

We thank the reviewer for this comment. It is the splenocytes used in panel A and CD8+ TILs for the rest of the panels (B-F). To avoid confusion we have now split the figures in new Sup Fig S12 (old S7 A) and new Sup Fig. S13 (Old S7B-F). We have updated the new figure legends with clear information.

13. Since CD19 is expressed by normal B cells, what about the impact on these cells *in vitro* and *in vivo* by the exposure/administration of CD19-targeted CAR-T and CAR(NAP)-T? Moreover at least one experiment by using as target cells unrelated and normal cells could be shown.

We appreciate the reviewer's comment. We observe that CAR(NAP)-T treated mice have a lower number of normal B-cells (CD19+ cells) in the blood, which is not observed for mice treated with conventional CAR-T (also included as Sup Fig S2F). This data is in line with observation reported by Pegram *et. al.* (PMID: 22354001) that conventional CAR-T requires chemotherapy pre-conditioning to achieve B-cell aplasia.

Regarding using irrelevant target cell as control. We think that the data presented in Supplementary Figure S1D, where we used both Daudi and BC-3 could explain the specificity, provided that we have always included a Mock-T control.

Reviewer #3 (Report for the authors (Required)):

The authors present a novel approach to enhancing CAR T cell efficacy through the additional introduction of the gene encoding H pylori neutrophil activating protein (NAP) which in turn allows the CAR T cell to introduce NAP to the tumor microenvironment leading to the recruitment of endogenous antitumor effectors including T cells as well as induction of DC maturation which in turn could enhance cross presentation of other tumor antigens to endogenous T cells. The authors demonstrate that these NAP secreting CAR T cells enhance tumor eradication in syngeneic tumor models and further demonstrate that these differences are due in part to modulation of the TME including DC maturation and recruitment of endogenous tumor specific T cells. The authors conclude that this is a proof of principle that bacterial derived proinflammatory factors may be a novel approach to enhance the efficacy of tumor targeted T cells in the context of adoptive cell therapies of cancer. The manuscript is well written and the data largely support the authors' conclusions. However, there are concerns regarding the approach and whether similar results have been

demonstrated using other proinflammatory molecules.

Critiques

1. A primary concern regarding this approach is the fact that similar outcomes have been demonstrated in the context of, for example, CAR T cells modified to secrete cytokines (i.e. IL-18), or express ligands which enhance DC maturation (i.e. CD40L). Both examples use non-immunogenic reagents. To this end, the work would seem to generate similar outcomes just using a different, but bacterial, proinflammatory molecule (NAP).

We thank the reviewer for this comment and fully agree that similar approaches of arming the CAR-T cells with host-derived factors such as IL12, IL18 or CD40L have been described by others. We fully acknowledge this in our manuscript. Even though we have not performed any head-to-head comparison, it is our believe that the use of a pluripotent bacterial-derived pathogenic factor would give a stronger immune stimulation than endogenous cytokines/chemokines. We have extended a discussion about this in our manuscript.

2. The authors note in the discussion that this approach may be limited due to the bacterial origin of NAP which quite reasonably would be immunogenic and in fact many patients may potentially already harbor neutralizing antibodies which could impair the efficacy of this approach. Have the authors immunized mice with NAP and then see if these CAR T cells still function well?

We greatly appreciate the concern raised by reviewer and have performed new experiment to address this question. We have summarized this in the section at the beginning of this response letter as all three reviewers had similar comments.

3. The authors have not fully studied the safety of this approach especially in the setting of syngeneic tumor models and in the context of systemic tumors. Post mortem analyses of treated mice to assess for systemic inflammation would be helpful. Further does NAP impact the murine immune effectors to a similar degree as in the context of a human immune system?

We appreciate the comment by the reviewer and have now performed new experiment to address the safety issue. In general, we did not see any extra treatment-related toxicity comparing conventional CAR-T and CAR(NAP)-T treatment, evidenced by analyzing a panel of blood cytokine levels related to cytokine release syndrome (one of the the most common side effects of CAR-T in humans) . We now also provide the body weight over time in all our mice models and report that we do not see any significant weight lost due to treatment. Both data sets are provided in new submitted version of manuscript as Supplementary Figure S9, S10.

We cannot guarantee that NAP has similar effect on the murine and human immune systems. However, the parameters that we have analyzed are similar between the murine and human immune response, including NAP-mediated DC recruitment and further activation of CAR(NAP)-T cells.

4. Have the authors looked into the long term persistence of the CAR(NAP) T cells? This is highly relevant given the clinical relevance of CAR T cell persistence and even more so if NAP induces T cell mediated immune responses which could lead to rapid elimination of the CAR T cells. This could readily be explored in the syngeneic mouse models wherein CAR T cell persistence could be

evaluated as well as assessment of whether endogenous T cells demonstrate cytotoxicity to CAR(NAP) T cells versus CAR T cells.

We thank the reviewer for this valuable comment that we fully agree upon. We have performed new experiment regarding persistence and toxicity. The data are presented as Supplementary Figure S7 and S8. Encouragingly, we did not observe a more rapid decrease of CAR(NAP)-T cells from the blood of mice compared to CAR-T cells. The situation may of course be different in humans why we will have to assess this in a phase I clinical trial.

5. Some of the presented data does not support the authors' conclusions including figure 1E wherein there is no statistical difference between the CAR-T and CAR(NAP) T cell treated mice in survival, and all long term surviving mice were resistant to tumor rechallange. It is also curious that NSX2 tumor models, the overall survival advantage is small and there are no long term surviving mice.

There is no statistical difference between the CAR-T and CAR(NAP)-T group. We thank the reviewer for this comment and have now updated our text in the newly submitted manuscript. We agree that the overall survival in NXS2 GD2 model is small, but we also appreciate that the difference is statistically significant. Comparing the different models used, it might be due to different efficacy of different CARs. It can also be noted that the expression of GD2 on NXS2 cells is relatively low *in vitro* and that *in vivo* expression indicate both a stronger and a weaker positive peak (Supplementary Figure S3A). Furthermore, it has been reported that GD2-directed CAR-T cells have tonic signaling and may therefore be less potent than CD19-directed CAR-T cells (PMID: 25939063).

In conclusion when taking all models into consideration, we are confident in reporting that NAP-armed CAR-T cells have demonstrated improved therapeutic efficacy in multiple models.